# Optimization by Parallel Quasi-Quantum Annealing with Gradient-Based Sampling

**Yuma Ichikawa**
Fujitsu Limited,
Department of Basic Science
University of Tokyo

**Yamato Arai**
Fujitsu Limited,
Department of Basic Science
University of Tokyo

## Abstract

Learning-based methods have gained attention as general-purpose solvers due to their ability to automatically learn problem-specific heuristics, reducing the need for manually crafted heuristics. However, these methods often face scalability challenges. To address these issues, the improved Sampling algorithm for Combinatorial Optimization (iSCO), using discrete Langevin dynamics, has been proposed, demonstrating better performance than several learning-based solvers. This study proposes a different approach that integrates gradient-based update through continuous relaxation, combined with Quasi-Quantum Annealing (**QQA**). QQA smoothly transitions the objective function, starting from a simple convex function, minimized at half-integral values, to the original objective function, where the relaxed variables are minimized only in the discrete space. Furthermore, we incorporate parallel run communication leveraging GPUs to enhance exploration capabilities and accelerate convergence. Numerical experiments demonstrate that our method is a competitive general-purpose solver, achieving performance comparable to iSCO and learning-based solvers across various benchmark problems. Notably, our method exhibits superior speed-quality trade-offs for large-scale instances compared to iSCO, learning-based solvers, commercial solvers, and specialized algorithms.

## 1 Introduction

Combinatorial optimization (CO) problems aim to find the optimal solution within a discrete space, a fundamental challenge in many real-world applications (Papadimitriou & Steiglitz, 1998; Crama, 1997). Most CO problems are NP-hard, making it challenging to solve large-scale problems within feasible computational time. As a result, significant effort has been devoted to developing algorithms that efficiently produce high-quality approximate solutions. Heuristic methods have been widely used to obtain approximate solutions, but they often require problem-specific knowledge. Accordingly, increasing efforts have been directed toward developing general-purpose solvers that can be applied to a broad range of problems to reduce the need for problem-specific insights.

Sampling-based approaches have been proposed, which treat CO problems as sampling problems. Simulated annealing (SA) (Kirkpatrick et al., 1983), a widely known technique, leverages local thermal fluctuations and updates (Metropolis et al., 1953; Hastings, 1970). Additionally, techniques such as tempered transitions (Neal, 1996) and exchange Monte Carlo algorithms (Hukushima & Nemoto, 1996) have shown strong performance in practical CO problems (Johnson et al., 1989; 1991; Earl & Deem, 2005). However, these methods depend on local updates, where only one dimension is updated at a time, and the update process across dimensions typically cannot be parallelized. As a result, these methods become computationally infeasible for large-scale CO problems.

Learning-based methods have recently gained recognition as general-purpose solvers for their ability to learn problem-specific heuristics automatically. This reduces the need for manually designed heuristics and leverages modern accelerators like GPUs and TPUs. Some learning-based methods rely on supervised data, which is often difficult to obtain (Li et al., 2018; Gasse et al., 2019; Gupta et al., 2020). Reinforcement learning (Khalil et al., 2017; Kool et al., 2018; Chen & Tian, 2019) and unsupervised learning approaches (Karalias & Loukas, 2020; Wang et al., 2022; Wang & Li,

2023) have broadened their scope of applications. However, these approaches encounter challenges when applied to out-of-distribution instances, limiting their flexibility in addressing a wide range of problem distributions. In response, unsupervised learning-based solvers that do not depend on training data have emerged, leveraging the ability of machine learning models to learn meaningful representations (Schuetz et al., 2022a;b; Ichikawa, 2023). These methods address CO problems by optimizing the weight parameters of machine learning models, but model selection is critical in determining solution quality Schuetz et al. (2023). Furthermore, additional computational costs arise because the number of learnable parameters exceeds the number of decision variables. Moreover, the performance of these methods compared to sampling-based solvers remains unclear.

On the other hand, the **i**mproved **S**ampling algorithm for **CO** (**iSCO**) (Sun et al., 2023b) was proposed, motivated by advances in Markov Chain Monte Carlo methods for discrete space. This method integrates discrete Langevin dynamics (Sun et al., 2023a) with traditional annealing techniques, demonstrating results comparable to or better than learning-based solvers by using gradient information from the objective function. This study proposes a different method that combines gradient-based updates through continuous relaxation with **Q**uasi **Q**uantum **A**nnealing (**QQA**), a technique inspired by quantum annealing (Kadowaki & Nishimori, 1998). QQA progressively transitions the objective function, beginning with a simple convex function minimized at half-integral values—similar to the state where the transverse field dominates in quantum annealing. In the end, this process converges to the original objective function, where the relaxed variables are minimized only within the discrete space—analogous to the classical state without the transverse field in quantum annealing. Unlike iSCO, this study introduces an extended Boltzmann distribution with a communication term between parallel runs. This distribution efficiently leverages GPUs, enabling broader exploration with minimal overhead and also accelerating convergence.

Numerical experiments on the same benchmark used as iSCO (Sun et al., 2023b), along with several additional benchmarks, demonstrate that our method is a competitive general-purpose solver, achieving performance comparable to iSCO and learning-based solvers across all benchmarks. For larger problems, our method provides better speed-quality trade-offs than iSCO, learning-based solvers, commercial solvers, and specialized algorithms.

## 2 BACKGROUND

**Combinatorial Optimization.** The goal of this study is to solve CO problems formulated as

$$\min_{\boldsymbol{x} \in \mathcal{X}(C)} f(\boldsymbol{x}; C), \quad \mathcal{X}(C) = \left\{ \boldsymbol{x} \in \{0,1\}^N \,\middle|\, \begin{array}{ll} g_i(\boldsymbol{x}; C) \leq 0, & \forall i \in [I], \\ h_j(\boldsymbol{x}; C) = 0 & \forall j \in [J] \end{array} \right\},$$

where $C \in \mathcal{C}$ denotes instance-specific parameters, such as a graph $G = (V, E)$, and $\mathcal{C}$ represents a set of all possible instances. The vector $\boldsymbol{x} = (x_i)_{1 \leq i \leq N} \in \{0,1\}^N$ represents the discrete decision variables, and $\mathcal{X}(C)$ denotes the feasible solution space. $f : \mathcal{X} \times \mathcal{C} \to \mathbb{R}$ is the cost function, while $g_i : \mathcal{X} \times \mathcal{C} \to \mathbb{R}$ and $h_j : \mathcal{X} \times \mathcal{C} \to \mathbb{R}$ represent inequality and equality constraints, respectively. We also use the shorthand notation $[N] = \{1, 2, \ldots, N\}$, with $N \in \mathbb{N}$.

To transform the constrained CO problem into a sampling problem, we reformulate it as an unconstrained CO problem using the penalty method:

$$\min_{\boldsymbol{x} \in \{0,1\}^N} l(\boldsymbol{x}; C, \boldsymbol{\lambda}), \quad l(\boldsymbol{x}; C, \boldsymbol{\lambda}) \triangleq f(\boldsymbol{x}; C) + \sum_{i=1}^{I+J} \lambda_i v_i(\boldsymbol{x}; C),$$

where $v_i : \{0,1\}^N \times \mathcal{C} \to \mathbb{R}, \forall i \in [I+J]$ represents penalty terms, which increase as the constraints are violated. For example, these penalty terms can be defined as

$$\forall i \in [I], \ v_i(\boldsymbol{x}; C) = \max(0, g_i(\boldsymbol{x}; C)), \quad \forall j \in [J], \ v_j(\boldsymbol{x}; C) = (h_j(\boldsymbol{x}; C))^2.$$

$\boldsymbol{\lambda} = (\lambda_i)_{1 \leq i \leq I+J} \in \mathbb{R}^{I+J}$ denotes the penalty parameters that control the balance between constraint satisfaction and cost function minimization.

**Energy Based Model.** For the penalized objective function $l(\boldsymbol{x}; C, \boldsymbol{\lambda})$, we define the following Boltzmann distribution:

$$P(\boldsymbol{x}; T) = \frac{1}{Z(T)} e^{-\frac{1}{T} l(\boldsymbol{x}; C, \boldsymbol{\lambda})}, \quad Z(T) = \sum_{x \in \mathcal{X}} e^{-\frac{1}{T} l(\boldsymbol{x}; C, \boldsymbol{\lambda})},$$

where $\sum_{\boldsymbol{x}}$ represents the sum over all possible values of $\boldsymbol{x}$, and $T \in \mathbb{R}_{\geq 0}$ is a parameter called temperature, which controls the smoothness of the distribution. As the temperature $T$ approaches infinity, the Boltzmann distribution $P(\boldsymbol{x}; T)$ converges to a uniform distribution over $\{0, 1\}^N$. When the temperature $T = 0$, the Boltzmann distribution becomes a uniform distribution over the optimal solutions of Eq. (2). The CO problem can be transformed into a sampling problem of $P(\boldsymbol{x}; T = 0)$.

**Metropolis–Hastings Algorithm.** The Metropolis–Hastings Algorithm (Metropolis et al., 1953; Hastings, 1970) is a general method for sampling from high-dimensional distributions. This algorithm uses a distribution $Q$ which satisfy the ergodicity condition. A Markov chain $(\boldsymbol{x}^{(t)})_{t=0}^T$ is constructed to have $P(\boldsymbol{x}; T)$ as its stationary distribution by following steps; starting from $\boldsymbol{x}^{(0)} \sim P^{(0)}$, where $P^{(0)}$ is the initial distribution, at each step $t$, propose $\boldsymbol{x}' \sim Q(\boldsymbol{x}'|\boldsymbol{x}^{(t)})$, and with probability

$$A(\boldsymbol{x}'|\boldsymbol{x}^{(t)}) = \min\left(1, \frac{P(\boldsymbol{x}'; T)Q(\boldsymbol{x}^{(t)}|\boldsymbol{x}')}{P(\boldsymbol{x}^{(t)}; T)Q(\boldsymbol{x}'|\boldsymbol{x}^{(t)})}\right)$$

accept $\boldsymbol{x}'$ as the next state $\boldsymbol{x}^{(t+1)} = \boldsymbol{x}'$, or with probability $1 - A(\boldsymbol{x}'|\boldsymbol{x}^{(t)})$, reject $\boldsymbol{x}'$ and retain the previous state $\boldsymbol{x}^{(t+1)} = \boldsymbol{x}^{(t)}$. The proposal distribution is typically a local proposal, such as a flipping a single bit. Gibbs sampling (Geman & Geman, 1984) is a local update method where the proposal distribution is the conditional distribution, $Q(\boldsymbol{x}^{(t)}|\boldsymbol{x}') = P(x_i^{(t)}|\boldsymbol{x}_{-i}; T)$, with $\boldsymbol{x}_{-i} = \boldsymbol{x} \setminus \{x_i\}$. In this case, the acceptance rate $A(\boldsymbol{x}'|\boldsymbol{x}^{(t)})$ is always 1. The effectiveness of the algorithm is highly influenced by the choice of the proposal distribution $Q(\boldsymbol{x}^{(t)}|\boldsymbol{x}')$.

**Simulated Annealing (SA).** SA was introduced by Kirkpatrick et al. (1983); Černỳ (1985). While the original CO problem can be solved by sampling from the distribution $\lim_{T \to 0} P(\boldsymbol{x}; T)$, directly sampling from the distribution is difficult due to its highly nonsmooth nature at low temperatures. To overcome, SA performs Metropolis–Hastings updates while gradually lowering the temperature $T$ to zero through a temperature path, $\mathcal{T} = (T_0, T_1, \ldots, T_M)$ where the path $T_0 > T_1 > \cdots > T_M \to 0$. This approach enables the system to reach a state with lower energy over time.

## 3 METHOD

We begin by explaining the continuous relaxation approach in Section 3.1. Section 3.2 introduces an extended Boltzmann distribution and QQA. Section 3.3 proposes the communication between parallel runs, and Section 3.4 summarizes optimization-specific techniques. This overall approach is termed Parallel Quasi-Quantum Annealing (**PQQA**).

### 3.1 CONTINUOUS RELAXATION STRATEGY

The continuous relaxation strategy reformulates a CO problem into a continuous one by converting discrete variables into continuous ones as follows:

$$\min_{\boldsymbol{p} \in [0,1]^N} \hat{l}(\boldsymbol{p}; C, \boldsymbol{\lambda}), \ \ \hat{l}(\boldsymbol{p}; C, \boldsymbol{\lambda}) \triangleq \hat{f}(\boldsymbol{p}; C) + \sum_{i=1}^{I+J} \lambda_i \hat{v}_i(\boldsymbol{p}; C),$$

where $\boldsymbol{p} = (p_i)_{1 \leq i \leq N} \in [0, 1]^N$ denotes the relaxed continuous variables, where each binary variable $x_i \in \{0, 1\}$ is relaxed to a continuous value $p_i \in [0, 1]$. $\hat{f} : [0, 1]^N \times \mathcal{C} \to \mathbb{R}$ represents the relaxation of $f$, satisfying $\hat{f}(\boldsymbol{x}; C) = f(\boldsymbol{x}; C)$ for $\boldsymbol{x} \in \{0, 1\}^N$. Similarly, for all $i \in [I + J]$, $\hat{v}_i$ denotes the relaxation of $v_i$, with $\hat{v}_i(\boldsymbol{x}; C) = v_i(\boldsymbol{x}; C)$ for $\boldsymbol{x} \in \{0, 1\}^N$. The landscape of relaxed objective function $\hat{l}(\boldsymbol{p}; C, \boldsymbol{\lambda})$ remains complex even after continuous relaxation, posing challenges for optimization. Moreover, continuous relaxation encounters rounding issues, where artificial rounding is needed to map the continuous back to the discrete solution, undermining the robustness.

### 3.2 QUASI-QUANTUM ANNEALING

We represent the relaxed variable $\boldsymbol{p} \in [0, 1]^N$ as a real-valued parameter $\boldsymbol{w} \in \mathbb{R}^N$ and a nonlinear mapping to evaluate gradients and introduce following entropy term to control the balance between

continuity and discreteness of relaxed variables:

$$\hat{r}(\sigma(\boldsymbol{w}); C, \boldsymbol{\lambda}, \gamma) = \hat{l}(\sigma(\boldsymbol{w}); C, \boldsymbol{\lambda}) + \gamma s(\sigma(\boldsymbol{w})),$$

where $\sigma : \mathbb{R}^N \to [0, 1]^N$ denotes element-wise mapping, such as the sigmoid function and $\gamma \in \mathbb{R}$ denotes a penalty parameter. The relaxed variables are represented as $\sigma(\boldsymbol{w}) = (\sigma(w_i))_{1 \le i \le N} \in [0, 1]^N$. The entropy term $s(\sigma(\boldsymbol{w}))$ is a convex function, which attains its minimum value of $0$ when $\sigma(\boldsymbol{w}) \in \{0, 1\}^N$ and its maximum when $\sigma(\boldsymbol{w}) = \mathbf{1}_N/2$. This study employs the following entropy:

$$s(\sigma(\boldsymbol{w})) = \sum_{i=1}^{N} \left\{ 1 - (2\sigma(w_i) - 1)^\alpha \right\}, \quad \alpha \in \{2n \mid n \in \mathbb{N}\},$$

which was introduced by Ichikawa (2023) and is referred to as $\alpha$-entropy. Furthermore, we extend the entropy term to general discrete optimization problems. Details of this generalization are provided in Appendix B. The effectiveness is demonstrated through numerical experiments on two benchmarks: the balanced graph partitioning in Section 5.4 and the graph coloring problem in Section 5.5.

Subsequently, the Boltzmann distribution in Eq. (2) is extended to the continuous space $[0, 1]^N$ using Eq. (3.2) as follows:

$$\hat{P}(\sigma(\boldsymbol{w}); \gamma, T) = \frac{1}{Z(\gamma, T)} e^{-\frac{1}{T}\hat{r}(\sigma(\boldsymbol{w}); C, \boldsymbol{\lambda}, \gamma)}, \quad Z(\gamma, T) = \int_{\mathbb{R}^N} d\boldsymbol{w} e^{-\frac{1}{T}\hat{r}(\sigma(\boldsymbol{w}); C, \boldsymbol{\lambda}, \gamma)}.$$

When $\gamma$ is negative, i.e., $\gamma < 0$, the relaxed variables tend to favor the half-integral value $\sigma(\boldsymbol{w}) = \mathbf{1}_N/2$, smoothing the non-convexity of the objective function $\hat{l}(\boldsymbol{p}; C, \boldsymbol{\lambda})$ due to the convexity of the entropy term $s(\boldsymbol{p})$. This state can be interpreted as a *quasi-quantum state*, where the values are uncertain between $0$ and $1$. Conversely, when $\gamma$ is positive, i.e., $\gamma > 0$, the relaxed variables favor discrete values. This state can be interpreted as a *classical state*, where the values are deterministically determined as either $0$ or $1$. Formally, the following theorem holds.

**Theorem 3.1.** *Under the assumption that the objective function $\hat{l}(\boldsymbol{p}; C)$ is bounded within the domain $[0, 1]^N$, the Boltzmann distribution $\lim_{\gamma \to +\infty} \lim_{T \to 0} \hat{P}(\sigma(\boldsymbol{w}); \gamma, T)$ converges to a uniform distribution over the optimal solutions of Eq. (2), i.e., $\boldsymbol{x}^* \in \arg\min_{\boldsymbol{x}} l(\boldsymbol{x}; C, \boldsymbol{\lambda})$. Additionally, $\lim_{\gamma \to -\infty} \lim_{T \to 0} \hat{P}(\sigma(\boldsymbol{w}); \gamma, T)$ converges to a single-peaked distribution $\prod_{i=1}^{N} \delta(\sigma(w_i) - 1/2)$.*

The detailed proof can be found in Appendix A.1.

The following gradient-based update can be used for any $\gamma$, provided that $r(\boldsymbol{w})$ is differentiable:

$$\boldsymbol{w}^{t+1} = \boldsymbol{w}^t - \eta \nabla_{\boldsymbol{w}} r(\sigma(\boldsymbol{w}^t); C, \boldsymbol{\lambda}) + \sqrt{2\eta T}\boldsymbol{\xi},$$

where $\eta$ is the time step size, and $\boldsymbol{\xi} \sim \mathcal{N}(\mathbf{0}_N, I_N)$ is Gaussian noise with $I_N \in \mathbb{R}^{N \times N}$ denoting an identity matrix and $\mathbf{0}_N$ representing the zero vector $(0, \dots, 0)^\top \in \mathbb{R}^N$. This update rule generates a Markov chain whose stationary distribution is given by Eq. (3.2), when $\eta$ is sufficiently small. Unlike the local updates in conventional SA for discrete problems, this method simultaneously updates multiple variables in a single step via the gradient, allowing for high scalability.

Following Ichikawa (2023), the annealing strategy is conducted on $\gamma$ from a negative to a positive value while updating the parameter $\boldsymbol{w}$. Initially, a negative $\gamma$ is set to enable extensive exploration by smoothing the non-convexity of $\hat{l}(\sigma(\boldsymbol{w}); C, \boldsymbol{\lambda})$, where the *quasi-quantum state* $\sigma(\boldsymbol{w}) = \mathbf{1}_N/2$ dominates. Subsequently, the penalty parameter $\gamma$ is gradually increased to a positive value until the entropy term approaches zero, i.e., $s(\sigma(\boldsymbol{w})) \approx 0$, to automatically round the relaxed variables by smoothing out continuous suboptimal solutions oscillating between $1$ and $0$. This annealing is similar to *quantum annealing*, where the system transitions from a state dominated by a superposition state of $0, 1$, induced by the transverse field, to the ground state of the target optimization problem. Therefore, we refer to this annealing as **Q**uasi-**Q**uantum **A**nnealing (**QQA**).

## 3.3 COMMUNICATION IN PARALLEL RUNS

Gradient-based updates in QQA enable efficient batch parallel computation for multiple initial values and instances $\{C_\mu\}_{\mu=1}^{P}$ using GPU or TPU resources, similar to those used in machine learning tasks. We propose communication between these parallel runs to conduct a more exhaustive search

and obtain diverse solutions. Specifically, we define the extended Boltzmann distribution over all $S$ parallel runs $\{\sigma(\boldsymbol{w}^{(s)})\}_{s=1}^S$ as follows:

$$\mathcal{P}\left(\{\sigma(\boldsymbol{w}^{(s)})\}_{s=1}^S; \gamma, T\right) \propto e^{-\frac{1}{T}\hat{R}\left(\{\sigma(\boldsymbol{w}^{(s)})\}_{s=1}^S; C, \boldsymbol{\lambda}, \gamma\right)},$$

$$\hat{R}\left(\{\boldsymbol{w}^{(s)}\}_{s=1}^S; C, \boldsymbol{\lambda}, \gamma\right) = \sum_{s=1}^S \hat{r}\left(\sigma(\boldsymbol{w}^{(s)}); C, \boldsymbol{\lambda}, \gamma\right) - S\alpha \sum_{i=1}^N \mathrm{STD}\left[\{\sigma(w_i^{(s)})\}_{s=1}^S\right],$$

where $\mathrm{STD}[\{a_k\}_{k=1}^K]$ denotes the empirical standard deviation $(\sum_{k=1}^K (a_k - \sum_{k'=1}^K a_{k'}/K)^2/K)^{1/2}$ and $\alpha \in [0,1]$ is a parameter controlling the diversity of the problem. A larger value of $\alpha$ enhances the exploration ability during the parallel run. The second term is a natural continuous relaxation of the Sum-Hamming distance (Ichikawa & Iwashita, 2024). As in Eq. (3.2), we perform Langevin updates on $\{\sigma(\boldsymbol{w}^{(s)})\}_{s=1}^S$ whose stationary distribution is given by Eq. (3.3).

### 3.4 Optimization-Specific Accelerations

We focus on optimization, rather than generating a sample sequence that strictly follows the probability distribution in Eq. (3.2). This section introduce several optimization-specific enhancements to the Langevin dynamics in Eq. (3.2) that are specifically designed to improve efficiency in optimization.

**Sensitive Transitions.** The update of $\boldsymbol{w}$ is linked with the objective function through the element-wise mapping $\sigma(\cdot)$, which can reduce sensitivity with respect to $\boldsymbol{w}$. To address this issue, we propose replacing the update in Eq. (3.2) with the following two-steps. In the first step, rather than updating $\boldsymbol{w}$ directly, we update the relaxation variables $\boldsymbol{p} \in \mathbb{R}^N$ as follows:

$$\boldsymbol{p}' = \boldsymbol{p}^t - \eta \nabla_{\boldsymbol{p}} \tilde{r}(\boldsymbol{p}^t; C, \lambda) + \sqrt{2\eta T}\boldsymbol{\xi},$$

where $\tilde{r} : \mathbb{R}^N \times \mathcal{C} \to \mathbb{R}$ is the relaxed form of $\hat{r}$, satisfying $\tilde{r}(\boldsymbol{p}; C) = \hat{r}(\boldsymbol{p}; C)$ for any $\boldsymbol{p} \in [0,1]^N$. This update leads to a more sensitive transition than updating $\boldsymbol{w}$, as it bypasses $\sigma(\cdot)$. However, $\boldsymbol{p}'$ may fall outside the range $[0,1]^N$ after the update Eq. (3.4). In the second step, we update $\boldsymbol{p}'$ by setting $\boldsymbol{p}^{t+1} = \sigma(\boldsymbol{p}')$, which ensures that $\boldsymbol{p}^{t+1}$ always lies within the range $[0,1]^N$.

Furthermore, the gradient-based update in Eq. (3.4) is replaced with more sophisticated optimizers to improve exploration efficiency. In this paper, we employ AdamW (Loshchilov & Hutter, 2017). Note that the stationary distribution of this optimization-specific update does not correspond to the Boltzmann distribution in Eq. (3.2).

## 4 Related Work

Sampling-based methods (Metropolis et al., 1953; Hastings, 1970; Neal, 1996; Hukushima & Nemoto, 1996) have been widely applied to various CO problems, including MIS (Angelini & Ricci-Tersenghi, 2019), TSP (Kirkpatrick et al., 1983; Černỳ, 1985; Wang et al., 2009), planning (Chen & Ke, 2004; Jwo et al., 1995), scheduling (Seçkiner & Kurt, 2007; Thompson & Dowsland, 1998), and routing (Tavakkoli-Moghaddam et al., 2007; Van Breedam, 1995). However, these methods generally depend on local updates, such as Gibbs sampling or single-bit flip Metropolis updates, which can restrict their scalability. To address this limitations, methods leveraging gradient information from the objective function have been proposed. Continuous relaxation techniques, conceptually related to our approach, have also been introduced (Zhang et al., 2012). However, these methods often struggle to capture the topological properties of discrete structures (Pakman & Paninski, 2013; Mohasel Afshar & Domke, 2015; Dinh et al., 2017; Nishimura et al., 2020). In contrast, our method gradually recovers the discrete nature of the original problem through QQA. Sun et al. (2023b) proposed iSCO, a sampling-based solver using discrete Langevin dynamics (Sun et al., 2023a; Zhang et al., 2022; Sun et al., 2021; Grathwohl et al., 2021), which accelerates traditional Gibbs sampling and improves performance. While iSCO approximates the acceptance rate of candidate transitions in discrete Langevin dynamics using a first-order Taylor expansion, its effectiveness for general discrete optimization problems remains uncertain. In addition to adjusting the temperature path, hyperparameter tuning is required for the path auxiliary sampler (Sun et al., 2021). Furthermore, communication between parallel chains in PQQA has not been implemented, preventing iSCO from fully utilizing the benefits of parallel execution on GPUs.

## 5 EXPERIMENT

In this section, the performance of PQQA is evaluated across five CO problems: maximum independent set (MIS), maximum clique, max cut, balanced graph partition, and graph coloring. For each problem, experiments are performed on synthetic and real-world benchmark datasets, including instances used in iSCO (Sun et al., 2023b), all benchmarks recently proposed (Sun et al., 2023a), and instances commonly used in learning-based solvers. Detailed formulations and instances of these problems are presented in Appendix D.1 and D.2. PQQA is compared with various baselines, including sampling-based methods, learning-based methods, heuristics, and both general and specialized optimization solvers. Notably, iSCO (Sun et al., 2023b) is employed as our direct baselines.

**Implementation.** We use the $\alpha$-entropy with $\alpha = 4$ across all experiments. The parameter $\gamma$ is increased linearly from $\gamma_{\min} = -2$ to $\gamma_{\max} = 0.1$ with each gradient update. After annealing, the relaxed variables are converted into discrete ones using the projection method: for all $i \in [N]$, we map as $x_i = \Theta(p_i - 1/2)$, where $\Theta$ is the step function. Notably, even with $\gamma_{\max} = 0.1$, the solution became almost binary, indicating robustness in the rounding process. We set $\sigma(\boldsymbol{w}) = \text{Clamp}(\boldsymbol{w})$, where $\text{Clamp}(\boldsymbol{w})$ constrains the values within the range $[0, 1]^N$. Specifically, for each $i \in [N]$, if $w_i$ is less than 0, it is set to 0; if $w_i$ exceeds 1, it is set to 1. For further discussion on why the sigmoid function is not applied for $\sigma(\boldsymbol{w})$, see Appendix C.1. The AdamW optimizer (Loshchilov & Hutter, 2017) is used. The parallel number $S$ is set to 100 or 1,000. We report the runtime of PQQA on a single V100 GPU. The runtime can be further improved with more powerful GPUs or additional GPUs. Section 5.6 provide an ablation study that examines the effect of different hyper-parameters. Refer to Appendix C for further implementation details. For all benchmark CO problems, the soft solution at the end of the training process became 0 or 1 within the 32-bit Floating Point range in PyTorch GPU. Additionally, no violations of the constraints were observed in our numerical experiments. Thus, following results presented in are feasible solutions.

**Evaluation Metric.** This experiment primarily evaluates solution quality using the approximation ratio (ApR), the ratio between the obtained and optimal solutions. For maximization problems, ApR $\leq 1$, and the solution is considered optimal when ApR $= 1$. In cases where an optimal solution is not guaranteed due to problem complexity, we directly compare objective functions or report the ratio relative to the solution found by a commercial solver with the best effort or asymptotic theoretical results. The specific definition of ApR used in each section is provided accordingly.

### 5.1 MAX INDEPENDENT SET

**SATLIB and Erdős–Rényi Graphs.** We evaluate PQQA using the MIS benchmarks from recent studies (Goshvadi et al., 2023; Qiu et al., 2022), which includes graphs from SATLIB (Hoos & Stützle, 2000) and Erdős–Rényi graphs (ERGs) of various sizes. Following Sun et al. (2023b), the instances consist of 500 SATLIB graphs, each containing with 403 to 449 clauses, corresponding at most 1,347 nodes and 5,978 edges, 128 ERGs with 700 to 800 nodes each, and 16 ERGs with 9,000 to 11,000 nodes each. PQQA is performed under four settings: a number of parallel runs with $S = 100$ or $S = 1000$ and fewer steps (3000 steps) or more steps (30000 steps), similar to the experiment in iSCO (Sun et al., 2023b). Table 1 shows the solution quality and runtime. The results indicate that PQQA achieves better speed-quality trade-offs than learning-based methods. In particular, PQQA outperforms both iSCO (Sun et al., 2023b) and KaMIS (Lamm et al., 2016; Hespe et al., 2019), the winner of PACE 2019 and a leading MIS solver, especially on both ERGs with shorter runtimes. Notably, for larger ERGs, PQQA finds much better solutions in a shorter time than iSCO and KaMIS.

**Regular Random Graphs.** We focus on MIS problems on regular random graphs (RRGs) with a degree $d$ greater than 16, which are known to be particularly difficult (Barbier et al., 2013). Previous studies have pointed out the difficulties in solving these instances with learning-based solvers (Angelini & Ricci-Tersenghi, 2023; Wang & Li, 2023). Building on the observations of Angelini & Ricci-Tersenghi (2023), we employ MIS problems on RRGs with $d = 100$ and $d = 20$ with sizes ranging from $10^4$ to $10^6$ variables. Additionally, we evaluate the ApR against the asymptotic theoretical values in the limit of an infinite number of variables (Barbier et al., 2013). If solvers cannot run even on V100 GPU, the result is reported as N/A. We employed QQA ($S = 1$) using two different step sizes: fewer steps (3,000 steps) and more steps (30,000 steps). Table 2 shows

Table 1: ApR and runtime are evaluated on three benchmarks provided by DIMES (Qiu et al., 2022). The ApRs are evaluated against the results obtained by KaMIS. Runtime is reported as the total clock time per instance in seconds (s/g) or minutes (m/g). The runtime and ApR are sourced from Sun et al. (2023b). Baselines include the OR solvers, learning-based methods using Reinforcement Learning (RL) and Supervised Learning (SL) combined with Tree Search (TS), Greedy decoding (G) or Sampling (S), and iSCO. Methods that fail to produce results within 10 times the time limit of DIMES are marked as N/A.

| Method | Type | SATLIB | | ER-[700-800] | | ER-[9000-11000] | |
|---|---|---|---|---|---|---|---|
| | | ApR↑ | Time↓ | ApR↑ | Time↓ | ApR↑ | Time↓ |
| KaMIS | OR | 1.000 | 4.50s/g | 1.000 | 24.44s/g | 1.000 | 28.5m/g |
| Gurobi | OR | 1.000 | 3.12s/g | 0.922 | 23.44s/g | N/A | N/A |
| Intel | SL+TS | N/A | N/A | 0.865 | 9.38s/g | N/A | N/A |
| | SL+G | 0.988 | 2.78s/g | 0.777 | 2.84s/g | 0.746 | 18.83s/g |
| DGL | SL+TS | N/A | N/A | 0.830 | 10.65s/g | N/A | N/A |
| LwD | RL+S | 0.991 | 2.26s/g | 0.918 | 2.97s/g | 0.907 | 28.35s/g |
| DIMES | RL+G | 0.989 | 2.90s/g | 0.852 | 2.87s/g | 0.841 | 19.54s/g |
| | RL+S | 0.994 | 2.43s/g | 0.937 | 5.63s/g | 0.873 | 46.91s/g |
| iSCO | fewer steps | 0.995 | 0.70s/g | 0.998 | 0.65s/g | 0.990 | 35.18s/g |
| | more steps | **0.996** | 1.83s/g | 1.006 | 2.61s/g | 1.008 | 4.69m/g |
| **PQQA** | fewer ($S = 100$) | 0.993 | 0.88s/g | 1.004 | 0.35s/g | 1.027 | 21.86s/g |
| | more ($S = 100$) | 0.994 | 8.71s/g | 1.005 | 3.33s/g | 1.039 | 3.66m/g |
| | fewer ($S = 1,000$) | **0.996** | 9.00s/g | 1.007 | 3.20s/g | 1.033 | 2.58m/g |
| | more ($S = 1,000$) | **0.996** | 1.50m/g | **1.009** | 32.06s/g | **1.043** | 25.50m/g |

Table 2: ApR and runtime are evaluated using five different seeds. ApR is measured relative to the asymptotic theoretical result (Barbier et al., 2013). Runtime is reported as the total clock time per instance in seconds (s/g) or minutes (m/g). The baselines include solvers such as the random greedy algorithm (GREEDY) (Angelini & Ricci-Tersenghi, 2019), CRA-GNN (Ichikawa, 2023), SA. Methods that failed to produce results on V100 GPU are marked as N/A.

| Method | | RRG ($d = 20$) | | | RRG ($d = 100$) | | |
|---|---|---|---|---|---|---|---|
| # nodes | | $10^4$ | $10^5$ | $10^6$ | $10^4$ | $10^5$ | $10^6$ |
| GREEDY | | 0.715 | 0.717 | 0.717 | 0.666 | 0.667 | 0.664 |
| | | (0.06s/g) | (9.76s/g) | (18.96m/g) | (0.02s/g) | (3.51s/g) | (5.16m/g) |
| CRA-GNN | | 0.922 | N/A | N/A | 0.911 | N/A | N/A |
| | | (3.483m/g) | | | (4.26m/g) | | |
| SA | fewer | 0.949 | 0.840 | 0.296 | 0.894 | 0.719 | 0.190 |
| | | (3.00m/g) | (3.00m/g) | (3.00m/g) | (3.00m/g) | (3.00m/g) | (3.00m/g) |
| | more | 0.971 | 0.943 | 0.695 | 0.926 | 0.887 | 0.194 |
| | | (30.00m/g) | (30.00m/g) | (30.00m/g) | (30.00m/g) | (30.00m/g) | (30.00m/g) |
| iSCO | fewer | 0.874 | 0.820 | 0.709 | 0.841 | 0.781 | 0.660 |
| | | (2.06s/g) | (3.75s/g) | (39.06s/g) | (2.04s/g) | (10.62s/g) | (1.36m/g) |
| | more | 0.956 | 0.923 | 0.895 | 0.916 | 0.884 | 0.850 |
| | | (11.11s/g) | (25.73s/g) | (6.27m/g) | (11.52s/g) | (1.60m/g) | (13.99m/g) |
| **PQQA** | fewer | 0.967 | 0.971 | 0.971 | 0.946 | 0.955 | 0.956 |
| | | (1.32s/g) | (1.35s/g) | (5.59s/g) | (1.34s/g) | (2.36s/g) | (18.47s/g) |
| | more | **0.976** | **0.980** | **0.980** | **0.957** | **0.966** | **0.966** |
| | | (3.77s/g) | (5.82s/g) | (47.99s/g) | (3.70s/g) | (15.69s/g) | (2.94m/g) |

the solution quality and runtime results across five different graphs generated with different random seeds. The results indicate that QQA performs well in these complex and large-scale instances. Notably, as the instance size increases, the performance gap between QQA and other methods grows, emphasizing the superior scalability for large-scale problems, which is a primary focus of heuristic approaches.

Table 3: ApR and runtime are evaluated on two benchmarks. Runtime is reported as the total clock time per instance in seconds (s/g) or minutes (m/g).

| Method | Twitter | RBtest |
|---|---|---|
| EPM (Karalias & Loukas, 2020) | $0.924 \pm 0.133$ (0.17s/g) | $0.788 \pm 0.065$ (0.23s/g) |
| AFF (Wang et al., 2022) | $0.926 \pm 0.113$ (0.17s/g) | $0.787 \pm 0.065$ (0.33s/g) |
| RUN-CSP (Toenshoff et al., 2021) | $0.987 \pm 0.063$ (0.39s/g) | $0.789 \pm 0.053$ (0.47s/g) |
| iSCO (Sun et al., 2023b) | $1.000 \pm 0.000$ (1.67s/g) | $0.857 \pm 0.062$ (1.67s/g) |
| **PQQA (Ours)** | $\mathbf{1.000 \pm 0.000}$ **(0.53s/g)** | $\mathbf{0.868 \pm 0.061}$ **(1.51s/g)** |

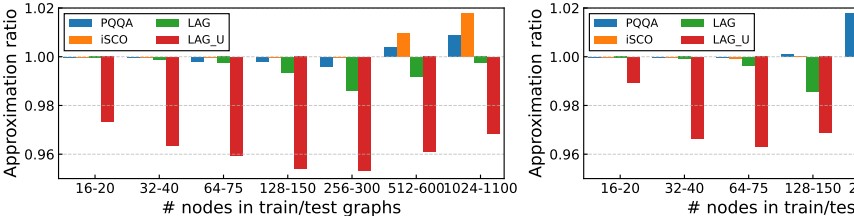

Figure 1: Approximation ratio comparison across different node sizes in train/test graphs.

## 5.2 MAX CLIQUE

Following Sun et al. (2023b), we present results on max clique benchmarks. We use the same instances from Karalias & Loukas (2020) and Wang & Li (2023), reporting the ApRs on synthetic graphs generated using the RB model (Xu et al., 2007) and a real-world Twitter graph (Jure, 2014). Table 3 shows the results of PQQA, which was run with 3,0000 steps and 1,000 parallel runs on each instance, compared to other learning-based methods trained on graphs from the same distribution. PQQA achieves significantly better solution quality while requiring only a minimal increase in runtime, considering that PQQA is executed without any prior training. Moreover, PQQA demonstrates better speed-quality tradeoffs than iSCO.

## 5.3 MAX CUT

We conduct max cut experiments following the same setup as in Sun et al. (2023b); Dai et al. (2021), where the benchmarks include random graphs and corresponding solutions obtained by running Gurobi for 1 hour. Specifically, the benchmarks includes both ERGs and Barabási–Albert (BA) graphs, with sizes ranging from 16 to 1,100 nodes and up to 91,239 edges. PQQA was run with 1,000 parallel processes and 30,000 steps for each instance. We report ApRs against the solutions provided by Gurobi and compare against the LAG (Dai et al., 2021) with either supervised learning-based approach (Li et al., 2018) denoted as LAG or unsupervised learning-based approach (Karalias

Table 4: Maxcut results on Optsicom.

| Method | ApR↑ |
|---|---|
| SDP | 0.526 |
| Approx | 0.780 |
| S2V-DQN | 0.978 |
| iSCO | **1.000** |
| **PQQA** | **1.000** |

& Loukas, 2020) denoted as LAG-U, and classical approach like semidefinite programming and approximated heuristics. Figure 1 shows that PQQA achieves optimal solutions in most cases and significantly outperforms Gurobi on large instances. PQQA performs comparably to iSCO. We also test PQQA on realistic instances (Khalil et al., 2017) which includes 10 graphs from the Optsicom project where edge weights are in $\{-1, 0, 1\}$. Table 4 shows the results show that PQQA can achieve the optimal solution in 1,000 steps, with a runtime of less than 5 second.

## 5.4 BALANCED GRAPH PARTITION

We next demonstrate the results of applying PQQA to general discrete variables, using the generalization of entropy detailed in Appendix B. Following Sun et al. (2023b); Nazi et al. (2019), PQQA is evaluated on balanced graph partition, including five different computation graphs from widely used deep neural networks. The largest graph, Inceptionv3 (Szegedy et al., 2017), consists of 27,144 nodes and 40,875 edges. The results are compared with iSCO, GAP (Nazi et al., 2019), which is a

Table 5: Balanceness are Graph partition are evaluated on 5 benchmarks.

| Metric | Methods | VGG | MNIST-conv | ResNet | AlexNet | Inception-v3 |
|--------|---------|-----|-----------|--------|---------|--------------|
| Edge cut ratio↓ | hMETIS | 0.05 | 0.05 | 0.04 | 0.05 | 0.04 |
| | GAP | **0.04** | 0.05 | 0.04 | **0.04** | 0.04 |
| | iSCO | 0.05 | **0.04** | 0.05 | 0.05 | 0.05 |
| | **PQQA** | **0.04** | **0.04** | **0.03** | **0.04** | **0.03** |
| Balanceness↑ | hMETIS | 0.99 | 0.99 | 0.99 | 0.99 | 0.99 |
| | GAP | 0.99 | 0.99 | 0.99 | 0.99 | 0.99 |
| | iSCO | 0.99 | 0.99 | 0.99 | 0.99 | 0.99 |
| | **PQQA** | 0.99 | 0.99 | 0.99 | 0.99 | 0.99 |

Table 6: Numerical results for COLOR graphs (Trick, 2002) are presented. For a specified number of colors, we report the cost, defined as the number of conflicts in the best coloring obtained by PQQA, PI-GCV, PI-SAGE (learning-based methods), and Tabucol (a tabu search-based method), sourced from Schuetz et al. (2022b); Yang et al. (2021).

| Graph | Colors | Tabucol | GNN | PI-GCN | PI-SAGE | **PQQA** |
|-------|--------|---------|-----|--------|---------|----------|
| anna | 11 | 0 | 1 | 1 | 0 | **0** |
| jean | 10 | 0 | 0 | 0 | 0 | **0** |
| myciel5 | 6 | 0 | 0 | 0 | 0 | **0** |
| myciel6 | 7 | 0 | 0 | 0 | 0 | **0** |
| queen5-5 | 5 | 0 | 0 | 0 | 0 | **0** |
| queen6-6 | 7 | 0 | 4 | 1 | 0 | **0** |
| queen7-7 | 7 | 0 | 15 | 8 | 0 | **0** |
| queen8-8 | 9 | 0 | 7 | 6 | 1 | **0** |
| queen9-9 | 10 | 0 | 13 | 13 | 1 | **0** |
| queen8-12 | 12 | 0 | 7 | 10 | 0 | **0** |
| queen11-11 | 11 | 20 | 33 | 37 | 17 | **11** |
| queen13-13 | 13 | 35 | 40 | 61 | 26 | **14** |

specialized learning architecture for graph partitioning, and hMETIS (Karypis & Kumar, 1999), a widely used framework for this problem. We use the edge cut ratio and balanceness for evaluation metrics, where a lower edge cut ratio indicates better performance, while a higher balanceness is preferable. Further details on these metrics and specific experimental conditions are provided in Appendix D.1. Table 5 shows that PQQA achieves better results with near-perfect balanceness and a lower cut ratio. Although PQQA required approximately 20 minutes for the largest graph, GAP, the fastest method, was completed in around 2 minutes. However, PQQA consistently achieved a lower edge cut ratio than GAP. Furthermore, with iSCO taking approximately 30 minutes to run, PQQA stands out as the faster approach, while still achieving a better edge cut ratio.

## 5.5 GRAPH COLORING

We evaluate PQQA on the graph coloring problem. Following the experimental setup of Schuetz et al. (2022b), we report the results on the publicly available COLOR dataset (Trick, 2002), commonly used in graph-based benchmark studies. For more detail on the dataset properties and specific experimental conditions, refer to Appendix D.1. The evaluation metric is the cost, representing the number of conflicts in the best coloring solution. We compare PQQA against PI-GCV and PI-SAGE (Schuetz et al., 2022b), both general-purpose unsupervised learning based solvers, and Tabucol (Yang et al., 2021), a tabu search-based heuristic that performs local search within a tabu list. As shown in Table 6 shows that PQQA achive the best results.

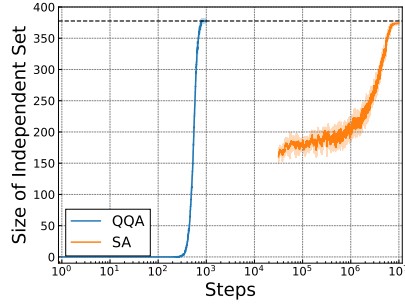

Figure 2: Comparison of SA and single-run QQA ($S = 1$).

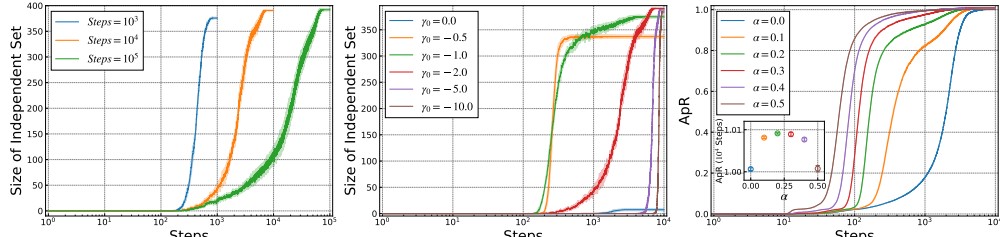

Figure 3: Ablation study on (left) annealing schedule speed, (middle) initial $\gamma$, and (right) communication strength $\alpha$. The shaded area denotes the standard deviation across five random seeds.

## 5.6 Ablation Study

**Detail Comparison with SA.** We compare our method to classical SA with Gibbs sampling. Figure 2 shows the ApR curves for an MIS on the largest ERGs in Section 5.1 as a function of the number of sampling steps. The results are reported as the mean and standard deviation across five random seeds. This experiment excluded communication between parallel runs to isolate the impact of gradient-based transitions, i.e., $S = 1$. The results demonstrate that QQA achieves a speedup of over $10^4$ times in the number of steps. Moreover, using gradient information improves the stability of the solution process.

**Schedule Speed and Initial $\gamma$.** We conduct an ablation study to evaluate the impact of different $\gamma$ schedules and initial values, $\gamma_0$. As in SA, QQA with a smaller initial value $\gamma_0$ and slower annealing achieves better results. Thus, we examine how the solution quality varies across different parameter settings. For the MIS on the largest ER graph in Section 5.1, Figure 3 (left) shows the independent set size across various annealing speeds, with the initial $\gamma_0$ fixed at $-2$. The annealing follows a linear schedule determined by the maximum number of steps, where fewer steps correspond to faster annealing. The results show no performance degradation, even with faster annealing. Indeed, QQA with $10^3$ steps maintains an ApR around 1.00, outperforming other learning-based solvers. Figure 3 (middle) shows the results for different $\gamma_0$ values under the same annealing speed, indicating that skipping the annealing phase when $\gamma < 0$ results in poor outcomes. Furthermore, the results are consistent when $\gamma_0$ is set below $-2$.

**Communication Strength.** An important contribution of this study is the introduction of communication between parallel chains, which was not discussed in iSCO (Sun et al., 2023b). Here, we conduct an ablation study on the communication strength, $\alpha$ in Eq. (3.3). The number of parallel chains is set to $S = 1,000$, and the performance is evaluated on the MIS on small ERGs, as described in Section 5.1 across various $\alpha$ values. Figure 3 shows the existence of the optimal $\alpha$ values, leading to the best ApR. Additionally, increasing $\alpha$ enhances convergence speed while maintaining performance comparable to the case of $\alpha = 0$. This improvement arises from the effect of the STD term in Eq. (3.3), which implicitly drives the relaxed variables toward 0 or 1. Additional theoretical insights are detailed in Appendix A.2.

## 6 Conclusion

PQQA, which integrates QQA, gradient-based updates, and parallel run communication, demonstrates performance comparable to or superior to iSCO and learning-based solvers across various CO problems. Notably, for larger problems, PQQA achieves a superior speed-quality trade-off. This suggests that future research on learning-based methods should carefully evaluate their efficiency compared to our GPU-based, general-purpose approach. Future work includes extending PQQA to the mixed-integer optimization and sampling tasks.

**Limitation.** Both sampling-based solvers, such as PQQA, iSCO, and SA, and UL-based solvers employ the penalty method in Eq. (2) to transform constrained CO problems into unconstrained ones. However, when even finding a single feasible solution is challenging, the formulation in Eq. (2) may fail to produce one. To alleviate this limitation, PQQA leverages large-scale parallel processing on GPUs to solve multiple instances with varying $\lambda$ values simultaneously.

## ACKNOWLEDGMENTS

We would like to thank Kai Nakaishi for valuable discussions. We also appreciate constructive feedback from anonymous reviewers and meta-reviewers. This work was supported by Fujitsu Limited. Additionally, YI was supported by the WINGS-FMSP program at the University of Tokyo.

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

# A DERIVATION AND ADDITIONAL THEORETICAL RESULTS

## A.1 DERIVATION OF THEOREM 3.1

In this section, we present the proof of Theorem 3.1.

**Theorem A.1.** *Under the assumption that the objective function $\hat{l}(\boldsymbol{p}; C)$ is bounded within the domain $[0,1]^N$, the Boltzmann distribution $\lim_{\gamma \to +\infty} \lim_{T \to 0} \hat{P}(\sigma(\boldsymbol{w}); \gamma, T)$ converges to a uniform distribution over the optimal solutions of Eq. (2), i.e., $\boldsymbol{x}^* \in \mathrm{argmin}_{\boldsymbol{x}} l(\boldsymbol{x}; C, \boldsymbol{\lambda})$. Additionally, $\lim_{\gamma \to -\infty} \lim_{T \to 0} \hat{P}(\sigma(\boldsymbol{w}); \gamma, T)$ converges to a single-peaked distribution $\prod_{i=1}^{N} \delta(\sigma(w_i) - 1/2)$.*

*Proof.* We begin by recalling the definition of $\hat{P}(\sigma(\boldsymbol{w}); \gamma, T)$:

$$\hat{P}(\sigma(\boldsymbol{w}); \gamma, T) = \frac{1}{Z(\gamma, T)} e^{-\frac{1}{T}\hat{r}(\sigma(\boldsymbol{w}); C, \boldsymbol{\lambda}, \gamma)}, \quad Z(\gamma, T) = \int \prod_{i=1}^{N} dw_i e^{-\frac{1}{T}\hat{r}(\sigma(\boldsymbol{w}); C, \boldsymbol{\lambda}, \gamma)}.$$

where $\hat{r}(\sigma(\boldsymbol{w}); C, \boldsymbol{\lambda}, \gamma)$ is defined as

$$\hat{r}(\sigma(\boldsymbol{w}); C, \boldsymbol{\lambda}, \gamma) = \hat{l}(\sigma(\boldsymbol{w}); C, \boldsymbol{\lambda}) + \gamma s(\sigma(\boldsymbol{w})),$$

where $\gamma \in \mathbb{R}$ and $s(\sigma(\boldsymbol{w}))$ is a convex function that takes its minimum value of 0 when $\sigma(\boldsymbol{w}) \in \{0,1\}^N$ and a maximum value when $\sigma(\boldsymbol{w}) = \mathbf{1}_N/2$. The set of global minimizers of $\hat{r}(\sigma(\boldsymbol{w}); C, \boldsymbol{\lambda}, \gamma)$ is defined as:

$$\Omega_\gamma^* = \mathrm{argmin}_{\boldsymbol{w} \in \mathbb{R}^N} \hat{r}(\sigma(\boldsymbol{w}); C, \boldsymbol{\lambda}, \gamma).$$

The remainder of the state space is denoted by $\Omega_\gamma = \mathbb{R}^N \setminus \Omega_\gamma^*$. In the limit as $T \to +0$, the following holds:

$$\lim_{T \to +0} \hat{P}(\sigma(\boldsymbol{w}); \gamma, T) = \frac{e^{-\frac{1}{T}\hat{r}(\sigma(\boldsymbol{w}); C, \boldsymbol{\lambda}, \gamma)}}{\int \prod_{i=1}^{N} dw_i e^{-\frac{1}{T}\hat{r}(\sigma(\boldsymbol{w}); C, \boldsymbol{\lambda}, \gamma)}},$$

$$= \lim_{T \to +0} \frac{e^{\frac{1}{T}(\hat{r}(\sigma(\boldsymbol{w}^*); C, \boldsymbol{\lambda}, \gamma) - \hat{r}(\sigma(\boldsymbol{w}); C, \boldsymbol{\lambda}, \gamma))}}{\int \prod_{i=1}^{N} dw_i e^{\frac{1}{T}(\hat{r}(\sigma(\boldsymbol{w}^*); C, \boldsymbol{\lambda}, \gamma) - \hat{r}(\sigma(\boldsymbol{w}); C, \boldsymbol{\lambda}, \gamma))}}$$

$$= \lim_{T \to +0} \frac{1}{\int \prod_{i=1}^{N} dw_i e^{\frac{1}{T}(\hat{r}(\sigma(\boldsymbol{w}^*); C, \boldsymbol{\lambda}, \gamma) - \hat{r}(\sigma(\boldsymbol{w}); C, \boldsymbol{\lambda}, \gamma))}} \delta(\boldsymbol{w} - \boldsymbol{w}^*)$$

$$+ \lim_{T \to +0} \frac{e^{\frac{1}{T}(\hat{r}(\sigma(\hat{\boldsymbol{w}}); C, \boldsymbol{\lambda}, \gamma) - \hat{r}(\sigma(\boldsymbol{w}); C, \boldsymbol{\lambda}, \gamma))}}{\int \prod_{i=1}^{N} dw_i e^{\frac{1}{T}(\hat{r}(\sigma(\hat{\boldsymbol{w}}); C, \boldsymbol{\lambda}, \gamma) - \hat{r}(\sigma(\boldsymbol{w}); C, \boldsymbol{\lambda}, \gamma))}} \delta(\boldsymbol{w} - \hat{\boldsymbol{w}})$$

$$= \lim_{T \to +0} \frac{1}{\int \prod_{i=1}^{N} dw_i \sum_{\boldsymbol{w}^* \in \Omega_\gamma^*} \delta(\boldsymbol{w} - \boldsymbol{w}^*)} \delta(\boldsymbol{w} - \boldsymbol{w}^*)$$

$$= \frac{1}{|\Omega_\gamma^*|} \delta(\boldsymbol{w} - \boldsymbol{w}^*),$$

where $\boldsymbol{w}^* \in \Omega_\gamma^*$ and $\hat{\boldsymbol{w}} \in \Omega_\gamma$. Here, we applied the property that $\lim_{T \to +0} e^{x/T} = 1$ when $x = 0$, and $\lim_{T \to +0} e^{x/T} = 0$ when $x < 0$. Therefore, in the limit as $T \to +0$, the distribution $\hat{P}(\sigma(\boldsymbol{w}); \gamma, T)$ converges to a uniform distribution over the set of global minimizers $\Omega_\gamma^*$ of Eq. (A.1). Given that $s(\sigma(\boldsymbol{w}))$ is a convex function with minimum value 0 when $\sigma(\boldsymbol{w}) \in \{0,1\}^N$ and maximum value at $1/2$, in the limit as $\gamma \to +\infty$, the entropy term $s(\sigma(\boldsymbol{w}))$ becomes dominant. Consequently, the state space is constrained to the set $\sigma(\boldsymbol{w}) \in \{0,1\}^N$ where $s(\sigma(\boldsymbol{w})) = 0$. Minimizing $\hat{l}(\sigma(\boldsymbol{w}); C, \boldsymbol{\lambda})$ within this constrained space yields:

$$\lim_{\gamma \to +\infty} \Omega_\gamma^* = \mathrm{argmin}_{\boldsymbol{x} \in \{0,1\}^N} l(\boldsymbol{x}; C, \boldsymbol{\lambda}).$$

Thus, in the limit as $\gamma \to \infty$ followed by $T \to +0$, $\hat{P}(\sigma(\boldsymbol{w}); \gamma, T)$ converges to a uniform distribution over the set of global minimizers of the discrete objective function $l(\boldsymbol{x}; C, \boldsymbol{\lambda})$. Conversely, as $\gamma \to -\infty$, the entropy $s(\sigma(\boldsymbol{w}))$ reaches its maximum value when $\sigma(\boldsymbol{w}) = \mathbf{1}_N/2$, leading to the minimization of $\hat{r}(\boldsymbol{w}; C, \boldsymbol{\lambda}, \gamma)$ at $\Omega_\gamma^* = \{\mathbf{1}_N/2\}$. Hence, we have:

$$\lim_{\gamma \to -\infty} \lim_{T \to +0} \hat{P}(\sigma(\boldsymbol{w}); \gamma, T) = \prod_{i=1}^{N} \delta(\sigma(w_i) - 1/2).$$

This concludes the proof. $\qquad\square$

## A.2 ADDITIONAL THEORETICAL RESULTS

The following proposition holds for the communication term in Eq. (3.3).

*Proposition* A.2. The function $\sum_{i=1}^{N} \mathbb{STD}[\{\sigma(w_i^{(s)})\}_{s=1}^{S}]$ is maximized when, for any $i \in [N]$, the set $\{\sigma(w_i^{(s)})\}_{s=1}^{S}$ consists of $S/2$ zeros and $S/2$ ones.

*Proof.* Consider the expression $\mathbb{STD}[\{\sigma(w^{(s)})\}_{s=1}^{S}]$, which can be expanded as:

$$\sum_{i=1}^{N} \mathbb{STD}[\{\sigma(w_i^{(s)})\}_{s=1}^{S}] = \sum_{i=1}^{N} \sqrt{\sum_{s=1}^{S} \left( \sigma(w_i^{(s)}) - \frac{1}{S} \sum_{s=1}^{S} \sigma(w_i^{(s)}) \right)^2}.$$

Thus, it suffices to solve the following maximization problem for any $i \in [N]$:

$$\max_{\{\sigma(w_i^{(s)})\}_{s=1}^{S}} \sum_{s=1}^{S} \left( \sigma(w_i^{(s)}) - \frac{1}{S} \sum_{s=1}^{S} \sigma(w_i^{(s)}) \right)^2.$$

This problem can be addressed using the method of Lagrange multipliers. Given that $\sigma(w^{(s)}) \in [0,1]^N$, we define the Lagrangian as:

$$\mathcal{L}(\{\sigma(w_i^{(s)}), \lambda_s, \nu_s\}_{s=1}^{S}) = \sum_{s=1}^{S} \left( \sigma(w_i^{(s)}) - \frac{1}{S} \sum_{s'=1}^{S} \sigma(w_i^{(s')}) \right)^2$$
$$+ \sum_{s=1}^{S} \lambda_s(-\sigma(w_i^{(s)})) + \sum_{s=1}^{S} \nu_s(1 - \sigma(w_i^{(s)})),$$

where $\lambda_s$ and $\nu_s$ are the Lagrange multipliers corresponding to the constraints $0 \leq \sigma(w_i^{(s)}) \leq 1$. The stationarity condition gives:

$$\frac{\partial \mathcal{L}(\{\sigma(w_i^{(s)}), \lambda_s, \nu_s\}_{s=1}^{S})}{\partial \sigma(w_i^{(s)})} = 2 \left( \sigma(w_i^{(s)}) - \frac{1}{S} \sum_{s'=1}^{S} \sigma(w_i^{(s')}) \right) \left( 1 - \frac{1}{S} \right)$$
$$- \frac{2}{S} \sum_{t \neq s} \left( \sigma(w_i^{(t)}) - \frac{1}{S} \sum_{s'=1}^{S} \sigma(w_i^{(s')}) \right) - \lambda_s + \nu_s$$
$$= 2 \left( \sigma(w_i^{(s)}) - \frac{1}{S} \sum_{s'=1}^{S} \sigma(w_i^{(s')}) \right) - \lambda_s + \nu_s = 0.$$

From dual feasibility, $\lambda_s \geq 0$ and $\nu_s \geq 0$ for any $s \in [S]$. Moreover, complementary slackness implies $\lambda_s \sigma(w_i^{(s)}) = 0$ and $\nu_s(\sigma(w_i^{(s)}) - 1) = 0$ for any $s \in [S]$. Considering the case where $\sigma(w_i^{(s)}) = 0$, $\lambda_s > 0$ due to complementary slackness, and hence $\sigma(w_i^{(s)}) = 0$. On the other hand, if $\sigma(w_i^{(s)}) = 1$, then $\nu_s > 0$, again due to complementary slackness, which forces $\sigma(w_i^{(s)}) = 1$. For the intermediate case where $0 < \sigma(w_i^{(s)}) < 1$, both $\lambda_s$ and $\nu_s$ must be zero, implying that $\sigma(w_i^{(s)}) = \sum_{s'} \sigma(w_i^{(s')})/S$. This means that all $\sigma(w_i^{(s)})$ are equal, leading to the variance being minimized to zero. To achieve the maximum variance, consider the scenario where $K$ out of $S$ variables are set to 0 and the remaining $S - K$ variables are set to 1. The variance is then computed as:

$$\sum_{s=1}^{S} \left( \sigma(w_i^{(s)}) - \frac{1}{S} \sum_{s=1}^{S} \sigma(w_i^{(s)}) \right)^2 = \frac{K}{S} \cdot \left( 0 - \frac{K}{S} \right)^2 + \frac{S - K}{S} \cdot \left( 1 - \frac{K}{S} \right)^2$$
$$= \frac{K(S - K)}{S^2}.$$

To maximize this quadratic function, differentiate with respect to $K$:

$$\frac{d}{dK} \left( \frac{K(S - K)}{S^2} \right) = \frac{S - 2K}{S^2}.$$

Setting this derivative to zero yields $K = {}^S\!/_2$. For even $S$, the maximum variance is:

$$\frac{K(S-K)}{S^2} = \frac{\left(\frac{S}{2}\right)\left(\frac{S}{2}\right)}{S^2} = \frac{1}{4}.$$

For odd $S$, $K$ is either $\lfloor {}^S\!/_2 \rfloor$ or $\lceil {}^S\!/_2 \rceil$. In both cases, the maximum variance is:

$$\frac{\lfloor \frac{S}{2} \rfloor \cdot \lceil \frac{S}{2} \rceil}{S^2} = \frac{(S-1)(S+1)}{4S^2} = \frac{S^2-1}{4S^2}.$$

As $S$ increases, the variance approaches $^1\!/_4$, consistent with the even case. $\qquad\square$

This result suggests that the communication term also favors binary values for $\sigma(\boldsymbol{w}) \in \{0,1\}^N$. Indeed, as shown in the ablation study in Figure 3 (right), increasing the strength of the communication term $\alpha$ accelerates convergence. This may be due to the communication term further reinforcing binary variables, in addition to the entropy term.

## B    GENERALIZATION TO DISCRETE OPTIMIZATION PROBLEMS

In this section, we generalize PQQA, defined for binary optimization, to a general discrete optimization problem. Specifically, we consider the following optimization problem:

$$\min_{\boldsymbol{x} \in \{1,\ldots,K\}^N} l(\boldsymbol{x}; C, \boldsymbol{\lambda}),$$

where $C$ represents the characteristic parameters of the problem, and $\lambda$ denotes the penalty coefficients. In this case, each discrete variable is relaxed to output probabilities for each discrete value, denoted as $\sigma(W)$, where $W \in \mathbb{R}^{N \times K}$ and the function $\sigma$ satisfies $\sum_{k=1}^{K} \sigma(W_{ik}) = 1$ for all $i \in [N]$, with the softmax function being a typical choice for $\sigma$. We then introduce an entropy term as follows:

$$\min_{W \in \mathbb{R}^{N \times K}} l(\sigma(W); C, \boldsymbol{\lambda}) + \gamma s_K(\sigma(W)),$$

where $\sigma(W)$ takes its maximum value of $^1\!/_K$ for all $i \in [N]$ and $k \in [K]$, and for all $i \in [N]$, there exists a $k$ such that $\sigma(W_{ik}) = 1$ and $\sigma(W_{ik'}) = 0$ for all $k' \in [K] \setminus \{k\}$. For example, we can consider the following entropy, which generalizes the $\alpha$-entropy:

$$s_K(\sigma(\boldsymbol{w})) = \sum_{i=1}^{N} \left\{ 1 - \frac{1}{(K-1)((K-1)^{\alpha-1}+1)} \sum_{k=1}^{K} (K\sigma(W_{ik})-1)^\alpha \right\}, \quad \alpha \in \{2n \mid n \in \mathbb{N}\}.$$

where $\gamma \in \mathbb{R}$ is a penalty parameter. This entropy has the following properties:
*Proposition* B.1. The entropy defined in Eq. (B) is equivalent with Eq. (3.2) when $K = 2$.

*Proof.* When $K = 2$, Eq. (B) can be written as

$$s_2(\sigma(W)) = \sum_{i=1}^{N} \left\{ 1 - \frac{1}{2}(2\sigma(W_{i1})-1)^\alpha - \frac{1}{2}(2\sigma(W_{i2})-1)^\alpha \right\}.$$

From the properties of $\sigma$, for any $i \in [N]$, we have $\sigma(W_{i1}) = 1 - \sigma(W_{i2})$. Using this relationship, we can rewrite the expression as follows:

$$s_2\left(\{\sigma(W_{i1})\}_{i=1}^{N}\right) = \sum_{i=1}^{N} \left\{ 1 - \frac{1+(-1)^\alpha}{2}(2\sigma(W_{i1})-1)^\alpha \right\}.$$

Finally, by noting that $\alpha \in \{2n \mid n \in \mathbb{N}\}$ and redefining $\boldsymbol{w} = (W_{i1})_{i=1}^{N}$, we obtain $s_2(\sigma(\boldsymbol{w})) = s(\sigma(\boldsymbol{w}))$. $\qquad\square$

Next, we generalize the following general discrete optimization problem:

$$\min_{\boldsymbol{x} \in \{\alpha_1,\ldots,\alpha_K\}} l(\boldsymbol{x}; C, \boldsymbol{\lambda}),$$

where $\boldsymbol{\alpha} = (\alpha_k)_{k=1}^{K} \in \mathbb{R}^K$. When applying PQQA to this problem, we use one-hot encoding to express the probability of $\alpha_k$ occurring for each $i \in [N]$ as $\sigma(W_{ik})$, and relax the discrete variable $x_i$ as $x_i = \boldsymbol{\alpha}^\top \sigma(W_{i:})$. To encode this with one-hot encoding, we also employ the entropy $s_K(\sigma(W))$ in Eq. (B). Moreover, PQQA can be generalized to mixed-integer optimization problems by adding the entropy term only to the integer variables.

## C    ADDITIONAL IMPLEMENTATION DETAILS

### C.1    REASONS FOR NOT USING THE SIGMOID FUNCTION

We provide an intuitive explanation for avoiding the use of the sigmoid function, $\sigma(x) = 1/(1+e^{-x})$ when mapping $w$ to the range $[0,1]$. Given the characteristics of the sigmoid function, $\boldsymbol{w}$ tends to take on extremely large positive or small negative values when $\sigma(\boldsymbol{w})$ is almost binary value. This makes it challenging for gradient-based methods to transition $w$ from very large positive values to very small negative values and vice versa once the variable approaches discrete values. Indeed, experimental results indicate that the sigmoid function performs less effectively than the clamp function. Further exploration of transformations tailored for CO beyond the clamp function remains for future research.

### C.2    ANNEALING SCHEDULE

We linearly increase $\gamma$ from $\gamma_{\min}$ to $\gamma_{\max}$ over the total steps. In general, we set $\gamma_{\min} = -2$ and $\gamma_{\max} = 0.1$. For the Max Cut problem, $\gamma_{\min} = -20$ was used for larger graphs, while $\gamma_{\min} = -5$ was applied for the remaining graphs.

### C.3    CONFIGURATION OF OPTIMIZER

The learning rate was set to an appropriate value from $\{1, 0.1, 0.01\}$, depending on the specific problem. The weight decay was fixed at $0.01$ and the temperature was fixed as $T = 0.001$.

### C.4    TRANSFORMATION FOR GENERAL DISCRETE VARIABLES

For discrete variables, each row $i \in [N]$ of the updated matrix $W \in \mathbb{R}^{N \times K}$ is transformed as follows:

$$\sigma(W_{ik}) = \frac{\mathrm{Clamp}(W_{ik})}{\sum_{k'=1}^{K} \mathrm{Clamp}(W_{ik'})}, \quad \forall k \in [K].$$

## D    ADDITIONAL EXPERIMENT DETAILS

### D.1    ENERGY FUNCTION OF BENCHMARK PROBLEMS

In this section we provide the actual energy function we used for each of the problems we experimented in the main paper. For a graph $G = (V, E)$ we label the nodes in $V$ from 1 to $N$. The adjacency matrix is represented as $A$. For a weighted graph we simply let $A_{ij}$ denote the edge weight between node $i$ and $j$. For constraint problems, we follow Sun et al. (2022) to select penalty coefficient $\lambda$ as the minimum value of $\lambda$ such that $x^* = \mathrm{argmin}_{x \in \{0,1\}^N} l(\boldsymbol{x}; \boldsymbol{\lambda}, C)$ is achieved at $\boldsymbol{x}^*$ satisfying the original constraints. Such a choice of the coefficient guarantees the target distribution converges to the optimal solution of the original CO problems while keeping the target distribution as smooth as possible.

**Max Independent Set.**    The MIS problem is a fundamental NP-hard problem (Karp, 2010) defined as follows. Given an undirected graph $G(V, E)$, an independent set (IS) is a subset of nodes $\mathcal{I} \in V$ where any two nodes in the set are not adjacent. The MIS problem attempts to find the largest IS, which is denoted $\mathcal{I}^*$. In this study, $\rho$ denotes the IS density, where $\rho = |\mathcal{I}|/|V|$. To formulate the problem, a binary variable $x_i$ is assigned to each node $i \in V$. Then the MIS problem is formulated as follows:

$$\min_{\boldsymbol{x} \in \{0,1\}^N} -\sum_{i=1}^{N} c_i x_i, \ \ \text{s.t.}\ x_i x_j = 0, \ \ \forall (i, j) \in E.$$

We use the corresponding energy function in the following quadratic form:

$$l(\boldsymbol{x}; A, \lambda) = -\boldsymbol{c}^\top \boldsymbol{x} + \lambda \frac{\boldsymbol{x}^\top A \boldsymbol{x}}{2}.$$

where the first term attempts to maximize the number of nodes assigned 1, and the second term penalizes the adjacent nodes marked 1 according to the penalty parameter $\lambda$. In our experiments

$c = \mathbf{1}_N$ and we use $\lambda = 2$. First, for every $d$, a specific value $\rho_d^*$, which is dependent on only the degree $d$, exists such that the independent set density $|\mathcal{I}^*|/|V|$ converges to $\rho_d^*$ with a high probability as $N$ approaches infinity (Bayati et al., 2010). Second, a statistical mechanical analysis provides the typical MIS density $\rho_d^{\text{Theory}}$ and we clarify that for $d > 16$, the solution space of $\mathcal{I}$ undergoes a clustering transition, which is associated with hardness in sampling (Barbier et al., 2013) because the clustering is likely to create relevant barriers that affect any algorithm searching for the MIS $\mathcal{I}^*$. Finally, the hardness is supported by analytical results in a large $d$ limit, which indicates that, while the maximum independent set density is known to have density $\rho_{d\to\infty}^* = 2\log(d)/d$, to the best of our knowledge, there is no known algorithm that can find an independent set density exceeding $\rho_{d\to\infty}^{\text{alg}} = \log(d)/d$ (Coja-Oghlan & Efthymiou, 2015).

**Max Clique.** The max clique problem is equivalent to MIS on the dual graph. The max clique the integer programming formulation as

$$\min_{\boldsymbol{x}\in\{0,1\}^N} -\sum_{i=1}^N c_i x_i, \ \ \text{s.t. } x_i x_j = 0, \ \ \forall (i,j) \notin E.$$

The energy function is expressed as

$$l(\boldsymbol{x}; A, \lambda) = -\boldsymbol{c}^\top \boldsymbol{x} + \frac{\lambda}{2}\left(\mathbf{1}_N^\top \boldsymbol{x}(\mathbf{1}_N^\top \boldsymbol{x} - 1) - \boldsymbol{x}^\top A \boldsymbol{x}\right),$$

where $\mathbf{1}_N$ denotes the vector $(1, \ldots, 1)^\top \in \mathbb{R}^N$. In our experiments $\boldsymbol{c} = \mathbf{1}_N$ and we use $\lambda = 2$.

**Max Cut.** The MaxCut problem is also a fundamental NP-hard problem (Karp, 2010) with practical application in machine scheduling (Alidaee et al., 1994), image recognition (Neven et al., 2008) and electronic circuit layout design (Deza & Laurent, 1994). Given an graph $G = (V, E)$, a cut set $\mathcal{C} \in E$ is defined as a subset of the edge set between the node sets dividing $(V_1, V_2 \mid V_1 \cup V_2 = V, \ V_1 \cap V_2 = \emptyset)$. The MaxCut problems aim to find the maximum cut set, denoted $\mathcal{C}^*$. Here, the cut ratio is defined as $\nu = |\mathcal{C}|/|\mathcal{V}|$, where $|\mathcal{C}|$ is the cardinality of the cut set. To formulate this problem, each node is assigned a binary variable, where $x_i = 1$ indicates that node $i$ belongs to $V_1$, and $x_i = 0$ indicates that the node belongs to $V_2$. Here, $x_i + x_j - 2x_i x_j = 1$ holds if the edge $(i, j) \in \mathcal{C}$. As a result, we obtain the following:

$$\min_{\boldsymbol{x}\in\{0,1\}^N} -\sum_{(i,j)\in E} A_{ij}\left(\frac{1 - (2x_i - 1)(2x_j - 1)}{2}\right)$$

Due to no constraints on this problem, the energy function can be expressed as

$$l(\boldsymbol{x}; A) = -\sum_{(i,j)\in E} A_{ij}\left(\frac{1 - (2x_i - 1)(2x_j - 1)}{2}\right).$$

**Balanced Graph Partition.** The balanced graph partition is formulated as follows:

$$\min_{\boldsymbol{x}\in\{0,1,\ldots,k\}^N} \sum_{s=1}^k \sum_{(i,j)\in E} \mathbb{I}\left[x_i \neq x_j \,\&\&\, (x_i = s \| x_j = s)\right] + \lambda \sum_{s=1}^k \left(\frac{d}{k} - \sum_{i=1}^N \mathbb{I}(x_i = s)\right)^2$$

where $k$ is the number of partitions. The goal of graph partitioning is to achieve balanced partitions while minimizing the edge cut. The quality of the resulting partitions is assessed using the following metrics: (1) Edge cut ratio, defined as the ratio of edges across partitions to the total number of edges, and (2) Balancedness, calculated as one minus the difference between the number of nodes in each partition and the ideal partition size as follows:

$$\text{Balanceness} = 1 - \sum_{s=1}^k \left(\frac{1}{k} - \frac{\sum_{i=1}^d \mathbb{I}(x_i = s)}{N}\right)^2.$$

Table 7: Synthetic data statistics.

| | MIS | | max clique | maxcut | |
|---|---|---|---|---|---|
| Name | ER-[700-800] | ER-[9000-11000] | RB | ER | BA |
| # max nodes | 800 | 10,915 | 475 | 1,100 | 1,100 |
| # max edges | 47,885 | 1,190,799 | 90,585 | 91,239 | 4,384 |
| # instances | 128 | 16 | 500 | 1,000 | 1,000 |

Table 8: Synthetic data statistics of graph coloring.

| Graph | # nodes | # edges | colors |
|---|---|---|---|
| anna | 138 | 493 | 11 |
| jean | 80 | 254 | 10 |
| myciel5 | 47 | 236 | 6 |
| myciel6 | 95 | 755 | 7 |
| queen5-5 | 25 | 160 | 5 |
| queen6-6 | 36 | 290 | 7 |
| queen7-7 | 49 | 476 | 7 |
| queen8-8 | 64 | 728 | 9 |
| queen9-9 | 81 | 1056 | 10 |
| queen8-12 | 96 | 1368 | 12 |
| queen11-11 | 121 | 1980 | 11 |
| queen13-13 | 169 | 3328 | 13 |

Table 9: Real-world data statistics.

| | MIS | Max Clique | Maxcut | Balanced Graph Partition | | | | |
|---|---|---|---|---|---|---|---|---|
| Name | Satlib | Twitter | Optsicom | Mnist | Vgg | Alexnet | Resnet | Inception |
| # max nodes | 1,347 | 247 | 125 | 414 | 1,325 | 798 | 20,586 | 27,114 |
| # max edges | 5,978 | 12,174 | 375 | 623 | 2,036 | 1,198 | 32,298 | 40,875 |
| # instances | 500 | 196 | 10 | 1 | 1 | 1 | 1 | 1 |

**Graph Coloring.** The graph coloring problem is formulated as follows:

$$\min_{\boldsymbol{x}\in\{0,1,\ldots,K\}^N} \left( - \sum_{(i,j)\in E} \mathbb{I}[x_i = x_j] \right)$$

where $K$ is a number of color. Following Schuetz et al. (2022b), we evaluate several benchmark instances from the COLOR dataset (Trick, 2002) for graph coloring. These instances can be categorized as follows:

(1) Book graphs: For a given work of literature, a graph is created with each node representing a character. Two nodes are connected by an edge if the corresponding characters encounter each other in the book.

(2) Myciel graphs: This family of graphs is based on the Mycielski transformation. The Myciel graphs are known to be difficult to solve because they are triangle free (clique number 2) but the coloring number increases in problem size.

(3) Queens graphs: This family of graphs is constructed as follows. Given an $n$ by $n$ chessboard, a queens graph is a graph made of $n^2$ nodes, each corresponding to a square of the board. Two nodes are then connected by an edge if the corresponding squares are in the same row, column, or diagonal. In other words, two nodes are adjacent if and only if queens placed on these two nodes can attack each other in a single move. In all cases, the maximum clique in the graph is no more than $n$, and the coloring value is lower-bounded by $n$.

Table 10: ApR (size of independent set) and runtime are evaluated on three benchmarks provided by DIMES (Qiu et al., 2022). The ApR is assessed relative to the results obtained by KaMIS. Runtime is reported as the total clock time, denoted in seconds (s), minutes (m), or hours (h). Methods that fail to produce results within 10 times the time limit of DIMES are marked as N/A.

| Method | Type | SATLIB | | ER-[700-800] | | ER-[9000-11000] | |
|---|---|---|---|---|---|---|---|
| | | ApR (size) | Time | ApR (size) | Time | ApR (size) | Time |
| KaMIS | OR | 1.000 (425.96) | 37.58m | 1.000 (44.87) | 52.13m | 1.000 (381.31) | 7.6h |
| Gurobi | OR | 1.000 (425.95) | 26.00m | 0.922 (41.38) | 50.00m | N/A | N/A |
| Intel (Li et al., 2018) | SL+TS | N/A | N/A | 0.865 (38.80) | 20.00m | N/A | N/A |
| | SL+G | 0.988 (420.66) | 23.05m | 0.777 (34.86) | 6.06m | 0.746 (284.63) | 5.02m |
| DGL (Böther et al., 2022) | SL+TS | N/A | N/A | 0.830 (37.26) | 22.71m | N/A | N/A |
| LwD (Ahn et al., 2020) | RL+S | 0.991 (422.22) | 18.83 | 0.918 (41.17) | 6.33m | 0.907 (345.88) | 7.56m |
| DIMES (Qiu et al., 2022) | RL+G | 0.989 (421.24) | 24.17m | 0.852 (38.24) | 6.12m | 0.841 (320.50) | 5.21m |
| | RL+S | 0.994 (423.28) | 20.26m | 0.937 (42.06) | 12.01m | 0.873 (332.80) | 12.51m |
| iSCO (Sun et al., 2023b) | fewer | 0.995 (423.66) | 5.85m | 0.998 (44.77) | 1.38m | 0.990 (377.5) | 9.38m |
| | more | 0.996 (424.16) | 15.27m | 1.006 (45.15) | 5.56m | 1.008 (384.20) | 1.25h |
| PQQA (Ours) | fewer | 0.993 (423.018) | 7.34m | 1.004 (44.91) | 44.72s | 1.027 (391.50) | 5.83m |
| | more | 0.994 (423.57) | 1.21h | 1.005 (45.11) | 7.10m | 1.039 (396.06) | 58.48m |
| | fewer | 0.996 (424.06) | 1.25h | 1.007 (45.20) | 6.82m | 1.033 (393.94) | 41.26m |
| | more | 0.996 (424.44) | 12.46h | 1.009 (45.29) | 1.14h | 1.043 (397.75) | 6.80h |

## D.2  BENCHMARK DETAILS

We present additional details on our experiments. First, Table 7 and Table 8 shows the statistics of the synthetic datasets, including the maximum number of nodes and edges per graph and the number of test instances. Table 9 provides the corresponding statistics for the real-world graphs.

# E  ADDITIONAL RESULTS

## E.1  MIS

In this section, we present additional information in Table 10, including references for each method and the sizes of the maximum independent sets obtained, alongside the MIS results discussed in Table 1.

