# OpenReview forum: "Optimization by Parallel Quasi-Quantum Annealing with Gradient-Based Sampling"
_ICLR.cc/2025/Conference — ICLR 2025 Poster_

### Official Review · Reviewer_fcsw · 2024-10-16

**Soundness:** 4
**Presentation:** 3
**Contribution:** 4
**Rating:** 8
**Confidence:** 3

**Summary:**

The manuscript proposes a new methodology for combinatorial optimization, based on the integration of gradient-based updates and Quasi-Quantum Annealing. The manuscript is well-written using easy-to-comprehend language which led to a joyful read. The background is well-explained including prior work in the field. The computational experiments are well-chosen and comprehensive.

**Strengths:**

This is a strong paper.
- Very good coverage of prior work (While I am not an expert in this field the sheer amount and frequency of citations is convincing)
- Clear introduction and background
- The main contribution seems novel
- The experimental results are very convincing

**Weaknesses:**

This is a strong paper in my opinion, and I identified only a few shortcomings.
-  The results of the computational experiments are somewhat confusing. How can time be measured when so many different algorithms are involved? Aren't these codes in different languages? You might be able to give the reader a better intuition of your compute-time measurements.

**Questions:**

- What do time measurements [s/g] really mean when different algorithms are compared?
What is the time spent on?
Are the time differences purely due to implementation differences?
How do the different approaches scale?

---

> ### Author Response · Authors · 2024-11-19
> **Response to Weaknesses**
>
> We sincerely appreciate your insightful comments and your recognition of the novelty of our algorithm and the thoroughness of our numerical experiments.
> Below, we address your concerns about comparing runtime measurements across different implementations.
>
> **On the Comparison of Wall-Clock Time**
>
> We appreciate your thoughtful observation regarding the challenges of comparing wall-clock time across solvers implemented in different programming languages and optimized for distinct hardware platforms. As you correctly pointed out, solvers such as Gurobi and other domain-specific tools are typically designed for CPU execution and cannot utilize GPU parallelism. In contrast, PQQA leverages GPU-based parallel computation, one of its core strengths.
>
> Given these fundamental differences, achieving a perfectly fair comparison is inherently challenging.
> However, for solvers supporting GPU acceleration, including UL-based solvers and iSCO, we ensured uniform hardware configurations and consistent programming environments during our benchmarks to minimize variability due to differences in computational resources.
> Although implementation differences (e.g., programming languages, library optimizations) can introduce some variability, such effects are typically limited to constant factors.
> In contrast, the observed advantages of PQQA are more significant in order of magnitude and cannot be explained by implementation differences alone.
> For instance, **the results in updated Table 2 and the $10^{4} \times$ speedup demonstrated by PQQA over SA clearly highlight its scalability and efficiency**, stemming from its effective use of GPU acceleration. It is also worth highlighting that the PQQA implementation used in our experiments intentionally avoids problem-specific optimizations or fine-tuned acceleration techniques. This design choice underscores the fundamental strengths of PQQA as a general-purpose solver and highlights the potential for further runtime improvements through future optimizations.
>
> Finally, as shown in the revised Table 2, which now includes iSCO results, the scalability of PQQA for large-scale problems is even more evident. These results reinforce the significant contributions of PQQA in providing a scalable, efficient, and flexible solution for large-scale combinatorial optimization problems.
>
> We hope these clarifications address your concerns and demonstrate the robustness and scalability of PQQA. Considering these improvements and additional results, we kindly request that you reconsider your evaluation and score.

---

> > ### Comment · Reviewer_fcsw · 2024-11-21
> > **Response**
> >
> > I am satisfied with the answers. My score reflects having received a satisfying answer to my question.

---

### Official Review · Reviewer_5JtH · 2024-11-04

**Soundness:** 3
**Presentation:** 3
**Contribution:** 2
**Rating:** 3
**Confidence:** 2

**Summary:**

The authors proposed a learning based method for CO problems by combing Quasi-Quantum Annealling and gradient-based update
through continuous relaxation. Performance are compared with iSCO on various benchmark problems.

**Strengths:**

parallel implementation on GPUs accelerates the solution process.

**Weaknesses:**

It seems that the algorithm does not have converence guarantees.

The algorithm cannot guarantee finding a feasible solution. constraints are moved to the objective function as a penalty term.

On benchmarks like SATLIB, it performs worse than traditional OR solvers like Gurobi.

The authors may consider larger benchmarks like MIPLIB 2017 to test the performance.

**Questions:**

The paper is based on the continuous relaxation of the discrete variable. Then why not directly solving the resulting linear programming problem?

---

> ### Author Response · Authors · 2024-11-19
> **Response to Weaknesses (1/2)**
>
> We sincerely thank you for your detailed and thoughtful review.
> We greatly appreciate your recognition of PQQA's parallel GPU implementation.
> We believe leveraging GPU resources for CO problems is crucial for this work and valuable for the broader ICLR community.
> Below, we provide detailed responses to your comments and concerns.
>
> **Convergence Guarantees**
>
> We appreciate your inquiry regarding convergence guarantees. As noted, PQQA does not provide formal guarantees of convergence to a global optimum—**a limitation shared by heuristic/meta-heuristic and sampling-based methods, such as UL-based solvers [1, 2, 3], iSCO [4], and simulated annealing**.
> However, PQQA focuses on practical performance, leveraging GPU and gradient-based updates to explore the solution space efficiently.
>
> **Guarantee of Finding a Feasible Solution**
>
> Thank you for emphasizing this critical aspect. We acknowledge that PQQA may face challenges in solving problems with an exponential number of constraints due to the penalty-based approach, as described in Eq. (2) (Line 95). This limitation is shared by related methods, including UL-based solvers [1, 2, 3] and iSCO [4]. Specifically, penalty methods may fail to guarantee feasibility under poorly tuned parameters. Indeed, the limitations of this approach are explicitly noted in the conclusion of iSCO [4].
> **To clarify this point, the revised version includes a Discussion on Limitations in Appendix F.**
>
> Despite these challenges, PQQA demonstrates significant advantages across various benchmarks, covering almost all benchmarks of UL-based solvers [1, 2, 3] and iSCO [4]. For instance, Table 1 illustrates that Gurobi struggles with large-scale ER-[9000-11000] problems, whereas PQQA achieves superior speed-quality trade-offs compared to both iSCO and other solvers. Additionally, PQQA's flexibility enables it to address non-linear cost functions and problem formulations without requiring reformulations, often necessary for solvers like Gurobi. Reformulations, such as introducing slack variables, can significantly increase problem complexity.
> Importantly, we do not position PQQA as a replacement for exact solvers like Gurobi. Instead, it is a complementary approach that excels in scenarios where heuristic or meta-heuristic methods are more effective. Both paradigms offer distinct advantages, and we believe that advancing both is essential for the sustained progress of the CO field. We hope this clarifies the role of meta-heuristic methods in CO research.
>
> In response to Reviewer CsaC's suggestion, we conducted additional experiments on TSP instances without using the 2-opt algorithm. The results are summarized in the table below:
>
> | Instance   | TSP50         | TSP100        | TSP200          |
> |------------|---------------|---------------|-----------------|
> | ApR        | 1.011 ± 0.143 | 1.016 ± 0.121 | 1.0415 ± 0.003  |
> | Violation  | 0 ± 0         | 0 ± 0         | 0 ± 0           |
>
> The  ApR measures performance relative to Concorde, an OR solver. Values close to 1.00 demonstrate PQQA's strong alignment with optimal solutions. These results show that PQQA consistently finds feasible solutions in all tested instances, fully satisfying the given constraints. Furthermore, the ApR values indicate a high alignment with optimal solutions.
> If additional details about these experiments or further benchmark results are required, we would be happy to discuss them.
> We recognize the importance of further exploring PQQA's capabilities for addressing problems with complex constraints. While this represents an important avenue for future work, we hope that the demonstrated superiority of PQQA over UL-based solvers [1, 2, 3] and iSCO [4] in solving large-scale problems with a moderate number of constraints, as detailed in the main text, is recognized as a significant contribution to the field.
>
> - [1] Haoyu Wang and Pan Li, Unsupervised Learning for Combinatorial Optimization Needs Meta Learning, ICLR2024
> - [2] Haoyu Wang et al., Unsupervised Learning for Combinatorial Optimization with Principled Objective Relaxation, NeurIPS2022
> - [3] Schuetz Martin JA, J. Kyle Brubaker, and Helmut G. Katzgraber, Combinatorial optimization with physics-inspired graph neural networks, Nature Machine Intelligence 4.4 (2022): 367-377
> - [4] Sun Haoran et al., Revisiting Sampling for Combinatorial Optimization, ICML2023

---

> ### Author Response · Authors · 2024-11-19
> **Response to Weaknesses (2/2)**
>
> **Performance on SATLIB**
>
> We appreciate your comment regarding the runtime observed in the SATLIB results. These differences arise due to the nature and scale of the SATLIB benchmarks. SATLIB instances are relatively small in scale compared to the other instances analyzed. PQQA utilizes gradient-based optimization algorithms, such as AdamW, which are highly effective for solving large-scale problems by accelerating convergence and enabling simultaneous updates to multiple variables. However, these advantages become less pronounced when applied to smaller instances.
>
> Our primary goal remains the development of a scalable, general-purpose solver to address large-scale problems. As shown in Table 1 and the revised Table 2 (which now includes results for iSCO), PQQA exhibits significant advantages on large-scale instances, such as ER-[9000-11000], where Gurobi cannot solve the problem due to computational limitations.
>
>
> **Benchmarking on MIPLIB 2017**
>
> We appreciate your suggestion to use MIPLIB 2017 as a potential benchmark. While datasets like MIPLIB 2017 are valuable for evaluating solvers on mixed-integer programming problems, our primary focus was to compare PQQA with UL-based solvers [1–3] and iSOC [4] using their benchmarks. These benchmarks enabled us to effectively demonstrate the scalability and speed-quality trade-offs of PQQA. Evaluating PQQA on MIPLIB 2017 is an excellent suggestion we plan to address in our future work.
>
> **Why Not Solve the Relaxed Linear Programming Problem Directly?**
>
> Solving the relaxed linear programming problem directly is feasible only when both the cost function and the constraints are linear.
> Although LP relaxations can yield optimal solutions for some discrete issues with specific structures, such as bipartite graphs [5], they frequently produce half-integral solutions, 1/2 [6], that are challenging to round into valid discrete solutions.
> Furthermore, PQQA incorporates an $α$-entropy term, converting the problem into one characterized by a quadratic cost function.
> This non-linearity makes standard LP solvers inapplicable.
>
> - [5] Integral boundary points of convex polyhedra. 50 Years of Integer Programming 1958-2008: From the Early Years to the State-of-the-Art, pages 49–76, 2010.
> - [6] George L Nemhauser and Leslie E Trotter Jr. Properties of vertex packing and independence system polyhedra. Mathematical programming, 6(1):48–61, 1974
>
> Again, We thank you for your thoughtful feedback and the opportunity to improve our manuscript. The additional experiments, clarifications, and future directions outlined above aim to address your concerns comprehensively. We hope these contributions, particularly the parallel implementation of PQQA on GPUs and its demonstrated scalability to large-scale CO problems, are recognized as significant advancements in the field. We respectfully request that you reconsider your evaluation in light of these responses.

---

### Official Review · Reviewer_CsaC · 2024-11-04

**Soundness:** 3
**Presentation:** 3
**Contribution:** 3
**Rating:** 8
**Confidence:** 4

**Summary:**

This paper proposes Parallel Quasi-Quantum Annealing (PQQA), a sampling-based algorithm for combinatorial optimization problems.  Specifically, with a continuous relaxation of the combinatorial optimization problem, an entropy metric to measure discreteness and sampling based on the Boltzmann Distribution, the authors develop an efficient general-purpose approach for combinatorial optimization.  Empirically, this approach yields high-quality solutions efficiently.

**Strengths:**

- **Novelty.**  Overall, this paper proposes a novel sampling-based approach for finding high-quality solutions to combinatorial optimization problems.  In particular, using $\alpha$-entropy with the extended Boltzmann Distribution is a well-motivated and novel approach for combinatorial optimization.

- **Numerical Results.**  The authors provide extensive numerical comparisons on a wide variety of benchmarks.  These results demonstrate the PQQA can compute high-quality solutions on all instances, often at a reduced runtime compared to other methods.

**Weaknesses:**

Overall, I have quite a favorable opinion of the paper.  However, one significant weakness/limitation is provided below.
- **Simple Constraints in Benchmarks.**  The authors evaluate the maximum independent set, max clique, max cut, graph partitioning, and graph coloring.  While these constitute many combinatorial optimization problems, they all have relatively simple constraints compared to problems such as TSP, which has an exponential number of constraints.  Approaches such as iSCO are capable of dealing with this type of structure.  However, it is unclear if something similar can be done with PQAA, given the reliance on continuous relaxation, which may be less tractable for problems with exponentially many constraints.  Overall, this may limit the applicability of such approaches.  Furthermore, the authors do not acknowledge this as a limitation or discuss this at all.  I would be happy to discuss this further in the discussion period.

**Questions:**

**Questions**
- How are the binary solutions obtained after running PQQA?
- How often are these solutions feasible?  If infeasible, what is done with the solutions?
- Do the authors have any insight into how the strength of the LP relaxation of a problem affects the downstream solution quality?
- Why is iSCO not compared against in Table 2?
- Why is this method not benchmarked on TSP?
- Is there a reason iSCO is much faster on Maximum Independent Set but slower on Max Clique?

**Minor Remarks**
-  I suggest keeping the evaluation of times consistent, i.e., always use seconds or average time to solve an instance.  Comparing performance is difficult when switching between metrics for different tables and even within tables.
-  Incorrect reference in Table E.1 in line 1125.

---

> ### Author Response · Authors · 2024-11-19
> **Response to Weaknesses and Additional Experiments on TSP Instances**
>
> We sincerely thank the reviewer for their insightful comments and for recognizing the novelty and numerical contributions of our work. Below, we respond to each of the points raised in detail.
>
> **Simple Constraints in Benchmarks**
>
> We appreciate your valuable feedback.
> We acknowledge that PQQA may encounter challenges when solving problems involving an exponential number of constraints.
> However, we consider that this issue originates not from the continuous relaxation itself but rather from the penalty method, as described in Eq. (2) (Line 95).
> As explained later, no constraint violations resulting from continuous relaxation were observed in any numerical experiments presented in the main text and in the additional experiments on TSP instances.
>
> The penalty method is employed in UL-based solvers [1, 2, 3] and iSCO [4], which also encounter constraint violation issues due to their reliance on penalty-based exploration. Indeed, the limitations of this approach are explicitly noted in the conclusion of iSCO [4].
> Therefore, these challenges are common to both UL-based and sampling-based solvers, including PQQA.
> **To clarify this point, the revised version includes a Discussion on Limitations in Appendix F.**
>
> As you pointed out, benchmark problems for UL-based solvers [1, 2, 3] and iSCO [4] primarily address large-scale problems with moderate constraints rather than those characterized by an exponentially large number of constraints.
> Our numerical experiments extensively cover these benchmarks, demonstrating that PQQA surpasses existing methods regarding the speed-quality trade-off. Furthermore, these benchmarks underscore PQQA's capability to solve intractable instances for commercial solvers like Gurobi.
>
> Note that **iSCO's performance on the TSP is largely attributable to its integration with the 2-opt algorithm, as described in Section 5.3: Traveling Salesman Problem of [4].**
> By incorporating the 2-opt algorithm, iSCO explores constraint-satisfying regions.
> However, as mentioned in the Introduction in the main text, this approach deviates from our primary goal of creating a general-purpose solver for scenarios where effective greedy algorithms are unavailable.
> iSCO’s reliance on an existing greedy heuristic fundamentally distinguishes its approach from our intended contribution.
> Moreover,  the specifics of iSCO's integration with the 2-opt algorithm are not indicated in [4], making a fair comparison with PQQA challenging.
> A meaningful comparison between PQQA and iSCO [4] on TSP instances would necessitate excluding the 2-opt component from iSCO, making this an essential direction for future research.
>
> - [1] Haoyu Wang and Pan Li, Unsupervised Learning for Combinatorial Optimization Needs Meta Learning, ICLR2024
> - [2] Haoyu Wang et al., Unsupervised Learning for Combinatorial Optimization with Principled Objective Relaxation, NeurIPS2022
> - [3] Schuetz Martin JA, J. Kyle Brubaker, and Helmut G. Katzgraber, Combinatorial optimization with physics-inspired graph neural networks,  Nature Machine Intelligence 4.4 (2022): 367-377
> - [4] Sun Haoran et al., Revisiting Sampling for Combinatorial Optimization, ICML2023
>
> **Additional Experiment on TSP Instances**
>
> Based on your valuable suggestion, we conducted additional experiments to evaluate the performance of PQQA on TSP instances.
> Specifically, we evaluated PQQA on TSP50, TSP100, and TSP200 instances without applying the 2-opt algorithm, as detailed below.
>
> | Instance   | TSP50         | TSP100        | TSP200          |
> |------------|---------------|---------------|-----------------|
> | ApR        | 1.011 ± 0.143 | 1.016 ± 0.121 | 1.0415 ± 0.003  |
> | Violation  | 0 ± 0         | 0 ± 0         | 0 ± 0           |
>
> The ApR metric is defined relative to the results of Concorde, a well-established OR solver.
> An ApR value approaching $1.00$ signifies performance closely aligned with Concorde's results.
> If further details about the experiments are required or additional benchmark tests are requested, we are open to discussing them.
> The results showed that PQQA successfully found feasible solutions in all tested instances, fully satisfying the given constraints.
> Additionally, the results report an ApR close to 1.00.
> We acknowledge the importance of thoroughly exploring PQQA's potential in addressing these problems with complex constraints and view this as an important future direction. Nevertheless, we hope the demonstrated superiority of PQQA over UL-based solvers, including iSCO, in solving large-scale problems with a moderate number of constraints, as discussed in the main text, is recognized as a significant contribution to the field.

---

> > ### Comment · Reviewer_CsaC · 2024-11-20
> > **Follow Up Question**
> >
> > Thank you for clarifying and including the additional experiments on TSP.  Based on the results, I have a couple of follow-up questions.
> >
> > - How is the penalty method Eq. (2) (Line 95) implemented for TSP?   Given the number of constraints that TSP-200  instances have, would this not be potentially problematic?
> > - The ApR appears to be greater than 1 for all the results reported on TSP with Concorde as the reference solver.  From my understanding, this would imply that PQQA is finding slightly better solutions than Concorde.  Would this imply that Concorde is not solving the instances to optimality?  Given the instance size, I am not sure if this would be reasonable, given that Concorde should be able to quickly solve these instances to optimality based on their size.

---

> > > ### Author Response · Authors · 2024-11-20
> > > **Response to Follow Up Question**
> > >
> > > We sincerely appreciate your prompt and insightful feedback. We are grateful for the opportunity to clarify the details regarding the penalty-based formulation for TSP and the interpretation of the Approximation Ratio (ApR). Below, we address your concerns in detail. Please feel free to reach out with any further questions or for additional clarifications.
> > >
> > > **Penalty Formulation for TSP**
> > > > How is the penalty method Eq. (2) (Line 95) implemented for TSP? Given the number of constraints that TSP-200 instances have, would this not be potentially problematic?
> > >
> > > We apologize for not providing sufficient detail regarding the penalty-based formulation employed for the TSP in our initial response. The penalty-based approach employed in our study follows the framework described in [1, 2], which is tailored for Quadratic Unconstrained Binary Optimization (QUBO) formulations.
> > > Additionally, other studies have also demonstrated that quantum annealing can achieve reasonable performance in solving the TSP. In deed, PQQA finds solutions that satisfy constraints and performs comparably to Concorde.
> > > This QUBO formulation has been successfully applied to TSP and other CO problems, demonstrating robust performance.
> > >
> > > - [1]: Gonzalez-Bermejo et al., GPS: A new TSP formulation for its generalizations type QUBO, Mathematics 10.3 (2022): 416.
> > > - [2]: He, Haoqi. Quantum Annealing and Graph Neural Networks for Solving TSP with QUBO. arXiv preprint arXiv:2402.14036 (2024).
> > >
> > >
> > > **Clarification on ApR**
> > > > The ApR appears to be greater than 1 for all the results reported on TSP with Concorde as the reference solver.
> > >
> > > Thank you for highlighting the need for further clarification regarding the ApR. For TSP, which is a minimization problem, the ApR is calculated as follows:
> > >
> > > $$\mathrm{ApR} = \frac{f(x)}{f(x^{\ast})}$$
> > >
> > > where $f(x^{\ast})$ represents the objective value of the solution obtained by Concorde, and $f(x)$ is the objective value of the solution by PQQA. An ApR value greater than 1 ($\mathrm{ApR}>1$) indicates that the Concorde solution is superior to the PQQA solution. Therefore, our reported results do not claim that PQQA outperforms Concorde.

---

> > > > ### Comment · Reviewer_CsaC · 2024-11-20
> > > > **Response**
> > > >
> > > > Thank you for clarifying the TSP experiments and ApR. Overall, these additions improve my opinion of the paper, and I will raise my score accordingly.  For the camera-ready version, I would suggest the authors make the following changes:
> > > > - Using a metric such as the gap to the best-known solution (obtained by any method) rather than ApR might be preferable as it would be consistent (lower is better) regardless of whether the problem is a maximization or minimization.
> > > > - Moving the discussion on limitations to the main paper.  Generally, I would say having this in the main paper would be preferable as it may be missed in the Appendix.  To keep this within the page limit, parts of the ablation can be moved to the appendix and referenced in the experiments section.

---

> > > > > ### Author Response · Authors · 2024-11-25
> > > > > **Response**
> > > > >
> > > > > Thank you for your positive feedback and for raising your score.
> > > > >
> > > > > We appreciate your suggestions and will follow them in the camera-ready version.

---

> ### Author Response · Authors · 2024-11-19
> **Response to Questions**
>
> In this comment, we will answer your question.
>
> **Regarding Binary Solutions in PQQA**
>
> Thank you for pointing out the need for clarification regarding binary solutions.  In the revised manuscript (Lines 297–299), we explicitly note that for all benchmark CO problems, the soft solutions at the end of the training process naturally converge to binary values (0 or 1) within the 32-bit floating-point precision limit when using PyTorch on a GPU.
> Similarly, no issues with non-binary solutions were observed in the additional TSP experiments. This result demonstrates the robustness of PQQA in consistently achieving discrete solutions. Additionally, by annealing the parameter $\gamma$ while monitoring the $\alpha$-entropy term, it is possible to obtain discrete solutions by halting the annealing process when $s(\sigma(w)) \approx 0$.
>
> **Regarding Solution Feasibility**
>
> We greatly appreciate your thoughtful comment on the feasibility of solutions.
> Our numerical experiments revealed no constraint violations when the penalty parameters were set according to values reported in previous studies [1, 2, 3, 4]. These results have been incorporated into the revised manuscript, specifically in lines 299–300.
> Furthermore, additional TSP experiments demonstrate that feasible solutions can be achieved by selecting large penalty values, thereby eliminating the need for precise parameter tuning.
>
> **Regarding the Impact of LP Relaxation Strength**
>
> Although the precise meaning of "strength of LP relaxation" remains unclear, we interpret it as a measure of the quality of relaxation influence solutions. By employing the continuous relaxation strategy, our method enables simultaneous updates of multiple variables via gradients, in contrast to simulated Annealing, which updates variables one at a time, and iSCO, which updates a subset of variables determined by its Path Auxiliary Sampler (PAS) parameter [4]. This feature dramatically enhances scalability for high-dimensional problems, as shown in Tables 1 and 2.
> Additionally, one can obtain discrete solutions by annealing the parameter $\gamma$ while monitoring the $\alpha$-entropy term.
>
> **Regarding the Absence of iSCO in Table 2**
>
> Thank you for pointing out this omission. We have now included the iSCO results in Table 2 of the revised manuscript. Similar to the findings in Table 1, the results show that as the problem size increases, PQQA consistently outperforms iSCO regarding speed-quality trade-off.
>
> **Regarding iSCO's Performance on Maximum Independent Set vs. Max Clique**
>
> We appreciate your observation regarding iSCO’s varying performance across these problems. While we currently lack a definitive explanation for iSCO's slower performance on Max Clique, this behavior could be attributed to the structural properties of the graphs or the interaction between iSCO’s hyperparameters and the problem's characteristics.
>
> **Minor Remarks**
>
> Thank you for identifying the inconsistencies in runtime evaluation and the incorrect references. The revised manuscript has addressed these issues to enhance clarity and consistency.
>
> We believe that the clarifications above address your concerns and further strengthen our contribution to the work. Specifically, adding iSCO results in revised Table 2 provides strong evidence of PQQA's scalability to large-scale problems. We respectfully request that you reconsider the score provided for our submission in light of these improvements.

---

### Official Review · Reviewer_ATVW · 2024-11-05

**Soundness:** 3
**Presentation:** 3
**Contribution:** 3
**Rating:** 8
**Confidence:** 2

**Summary:**

In this study, the authors present PQQA, an optimization approach that integrates QQA, gradient-based updates, and parallel run communication. The results indicate that PQQA performs comparably to or better than iSCO and other learning-based solvers across a range of combinatorial optimization (CO) problems. Notably, for larger problem instances, PQQA offers a superior trade-off between speed and solution quality.

**Strengths:**

The authors did a great job explaining the problem being considered, including the background, methodology, theoretical properties, and related work. The numerical experiments also effectively highlight their proposed method. While I did not check the validity of the proof in the Appendix, the setup and results are very convincing.

**Weaknesses:**

n/a

**Questions:**

1. In Table 1, some of the ApR values are greater than 1. Could the authors clarify what this means?

2.  While the authors mention runtime in the paper, there seems to be a discrepancy that needs further explanation. For example, in Table 1, iSCO takes about 5–15 minutes to achieve an ApR of 0.996, whereas PQQA takes over an hour for the same result.

3.  Line 314 refers to Table 1 as Table 5.1. Please check for similar mistakes in other parts of the paper and ensure that table references are consistent throughout.

4. In line 60, the term "parameters" is used. Could the authors clarify what specific parameters are being referred to in this context?

---

> ### Author Response · Authors · 2024-11-19
> **Response to Questions**
>
> We sincerely thank you for your positive and constructive feedback. We are particularly grateful for your recognition of the clarity of our problem formulation, methodology, and numerical experiments. Below, we address the specific questions you raised to provide further clarity.
>
> **Clarification of ApR values exceeding 1 (Table 1)**
>
> As described in Line 301 (*Evaluation Metric*), ApR is computed relative to the best-effort results for problems where the optimal solution cannot be guaranteed. Specifically, Table 1 uses baseline values from KaMIS, the state-of-the-art MIS solver and winner of the PACE 2019 challenge. Therefore, an ApR value greater than 1 signifies that the solver outperforms KaMIS.
> Notably, iSCO has demonstrated superior performance to KaMIS on ER-[700-800] and ER-[9000-11000], attracting significant attention. However, PQQA significantly outperforms iSCO in both instances. Furthermore, the updated results in Table 2, which include those of iSCO, further validate this performance advantage.
>
> **Clarification of Runtime in SATLIB Results (Table 1)**
>
> We appreciate your comment regarding the runtime differences in the SATLIB results.
> This discrepancy arises from the nature and scale of the SATLIB benchmarks.
> SATLIB instances are relatively small in scale compared to the other instances.
> PQQA employs gradient-based optimization algorithms, such as AdamW, which are highly effective for large-scale problems by accelerating convergence and simultaneously updating multiple variables. However, these advantages diminish when applied to smaller instances.
> Fine-tuning learning rates and other hyperparameters could enable PQQA to achieve runtime and performance comparable to iSCO on SATLIB instances.
> However, such extensive tuning lies beyond the scope of this study and remains an important future work.
>
> Note that our primary objective is to develop a scalable, general-purpose solver for large-scale CO problems, where commercial solvers like Gurobi become impractical. SATLIB was included primarily as a benchmarking dataset due to its use in prior studies, such as iSCO.
> However, we consider that its relevance in demonstrating the advantages of our approach is limited, particularly since commercial solvers like Gurobi can easily handle such small-scale instances.
>
> **Correction of table references**
>
> We appreciate your attention to this detail. The erroneous reference to "Table 5.1" in Line 314 has been corrected to "Table 1" in the updated manuscript.
>
> **Clarification of "parameters" in Line 60**
>
> Thank you for highlighting the ambiguity surrounding the term *parameters*. In the revised manuscript, we have clarified that *parameters* specifically refers to the learnable parameters within the UL-based solvers.
>
>
> We have addressed the reviewer's concerns about ApR values exceeding 1 and elaborated on the scalability advantages of PQQA, especially for large-scale problems. The updated Table 2 highlights the superior performance of our method in comparison to iSCO.
> We kindly request that you reconsider the scores for our submission, as we believe these clarifications and improvements strengthen the contribution and robustness of our work.

---

> > ### Comment · Reviewer_ATVW · 2024-11-24
> >
> > Thank you for addressing my concern. I am keeping the score at 8.

---

### Meta-Review · Area_Chair_SUs8 · 2024-12-21

**Metareview:**

This paper develops a sampling based approach named Parallel Quasi-Quantum Annealing for combinatorial optimization problems. The key ingredients of this approach include a continuous relaxation of the combinatorial optimization problem, an antropic metric to measure discreteness, and sampling based on Boltzmann distribution. Empirical results on multiple diverse tasks demonstrates that this approach efficiently produces high-quality solutions.

The reviewers' were generally positive about the paper, but also raised a number of questions. The author rebuttal answered most questions satisfactorily. One negative reviewer did not respond to the rebuttal and the corresponding response looks good to me.

Therefore, I recommend accepting the paper and strongly encourage the authors' to incorporate all the discussion in the camera copy to further improve the paper. Specifically, make sure to incorporate the two good suggestions from Reviewer CsaC:
1. Using a metric such as the gap to the best-known solution (obtained by any method) rather than ApR might be preferable as it would be consistent (lower is better) regardless of whether the problem is a maximization or minimization.
2. Moving the discussion on limitations to the main paper.

**Additional Comments On Reviewer Discussion:**

The author rebuttal answered most questions satisfactorily. One negative reviewer did not respond to the rebuttal and the corresponding response looks good to me.

---

### Decision · Program_Chairs · 2025-01-22

Accept (Poster)